# High-performance shape-engineerable thermoelectric painting

Sung Hoon Park[1], Seungki Jo[1], Beomjin Kwon[2], Fredrick Kim[1], Hyeong Woo Ban[1], Ji Eun Lee[3], Da Hwi Gu[1], Se Hwa Lee[1], Younghun Hwang[1], Jin-Sang Kim[2], Dow-Bin Hyun[2], Sukbin Lee[1], Kyoung Jin Choi[1], Wook Jo[1] & Jae Sung Son[1]

Output power of thermoelectric generators depends on device engineering minimizing heat loss as well as inherent material properties. However, the device engineering has been largely neglected due to the limited flat or angular shape of devices. Considering that the surface of most heat sources where these planar devices are attached is curved, a considerable amount of heat loss is inevitable. To address this issue, here, we present the shape-engineerable thermoelectric painting, geometrically compatible to surfaces of any shape. We prepared $Bi_2Te_3$-based inorganic paints using the molecular $Sb_2Te_3$ chalcogenidometalate as a sintering aid for thermoelectric particles, with $ZT$ values of 0.67 for n-type and 1.21 for p-type painted materials that compete the bulk values. Devices directly brush-painted onto curved surfaces produced the high output power of $4.0 \, mW \, cm^{-2}$. This approach paves the way to designing materials and devices that can be easily transferred to other applications.

[1] School of Materials Science and Engineering, Ulsan National Institute of Science and Technology (UNIST), Ulsan 44919, Republic of Korea. [2] Center for Electronic Materials, Korea Institute of Science and Technology (KIST), Seoul 02792, Republic of Korea. [3] Thermoelectric Conversion Research Center, Korea Electrotechnology Research Institute, Changwon 51543, Republic of Korea. Correspondence and requests for materials should be addressed to J.S.S. (email: jsson@unist.ac.kr).

The thermoelectric (TE) effect has attracted considerable attention from various research areas, as its ability to directly convert between thermal and electrical energy offers a unique solution to sustainable power generation from waste heat sources[1–3]. The overall power generating performance of solid-state TE devices largely depends on the characteristics of the TE materials itself. This means that the efficiency can be estimated from a dimensionless figure-of-merit inherent in the materials: $ZT = (S^2\sigma T/\kappa)$, where $S$, $\sigma$, $\kappa$ and $T$ are the Seebeck coefficient, electrical conductivity, thermal conductivity and temperature, respectively. TE materials chipped into devices are mostly prepared in the form of cube or cuboid blocks from a TE ingot by means of a top–down dicing process[4–6]. A potential problem of this conventional procedure is a relatively high production cost due to the energy intensive processing for ingots such as zone-melting[7] or hot-pressing[8,9] as well as the post-processing due to shape control[10–12]. The latter, in fact, suffers from another problem that attempts to realize any complicated shape other than a cube are technically impossible within the context of mass production.

On the other hand, in the real-world applications, the minimization of heat loss due to incomplete contact between the surface of the heat source and the TE module is no less important than the figure-of-merit of materials[4,11–13]. It is noted that the majority of heat sources for TE generators has irregular shapes, where the conventional planar-structured TE devices composed of cubic blocks should fail in achieving a desirable contact (Fig. 1a,b).

One readily available solution to settling down the aforementioned issues would be to secure a way to maximizing the flexibility in the shape and dimension control of TE materials during the forming stage, where the well-established printing technology would best-serve the purpose[14–23]. However, this printing-based technology has faced at least two major challenges. One is the poor functional properties due to the unavoidable organic-conducting binders in the inks for electrical interconnection among TE particles at the expense of TE properties[20–22]. Although the properties can be enhanced by high temperature processing instead of using organic binders, such enhancement is quite limited; for example, the state of the art utilizing a printing technique for TE materials deposited on a glass fabric achieved $ZT$ values of 0.35 (n-type) and 0.27 (p-type)[23] which are at most 20–40% of the commonly reported values from the conventional processing[2] even with the sintering temperature of as high as 530 °C close to the melting point of the TE materials. The other is the limited choice of substrates, that is, the usual printing technique forces one to deposit TE materials only on a flat surface, though the targeted heat sources where TE modules are attached are generally curved[24].

As a solution to these challenges, we present the development of high-performance shape-engineerable TE painting via the molecule-level sintering effect[25,26]. To this end, we utilized the molecular $Sb_2Te_3$-based chalcogenidometalate (ChaM)[27–30] for n-type BiTeSe and p-type BiSbTe TE particles, which are arguably known as the best TE materials at near room temperature[31,32]. The $Sb_2Te_3$ ChaM turned out to promote the sintering process effectively even at as low as 350 °C without any remnant secondary phase, the presence of which jeopardizes the expected TE properties. With the processing optimized, we have achieved the $ZT$ values of 1.21 for p-type and 0.67 for n-type TE materials that are comparable to the bulk values[2] and three times higher than the best values among the printed TE materials reported in the literature[23]. To show the feasibility of the currently proposed technology, we fabricated TE generators through painting TE paints on flat, curved and large-sized hemispherical substrates, demonstrating that it is the most effective means of heat energy collection from any heat sources with exceedingly high output power density of 4.0 mW cm$^{-2}$, which is the best value among the reported printed TE generators[33].

## Results

The overall process for the TE painting in the current study is shown in Fig. 2. The prepared TE paints with the $Sb_2Te_3$ ChaM sintering aid are painted on a curved substrate and sintered at elevated temperatures (Fig. 2a–c), eventually, producing the curved painted TE device exhibiting high power generating performances (Fig. 2d). The resulting n- and p-type painted materials in the TE generators exhibit the $ZT$ values competing those of bulk $Bi_2Te_3$-based materials (Fig. 2e). The detailed results and the related discussion on the each steps are described below.

**$Bi_2Te_3$-based inorganic TE paints.** The $Sb_2Te_3$-based ChaM was synthesized by dissolving bulk elemental Sb and Te in a thiol–diamine mixture[34–36] instead of the widely used a $N_2H_4$ solvent[25–28,37–41] due to its high-level toxicity. Sb $3d^{3/2}$, Sb $3d^{5/2}$ and Te $3d^{5/2}$ peaks corresponding to metallic bonding peaks are identified in the X-ray photoelectron spectra (Supplementary Fig. 1), indicating the formation of an ionic $Sb_2Te_4$ phase from elements[42]. We observed that this soluble compound decomposes into rhombohedral $Sb_2Te_3$ and hexagonal Te on mild heat treatment above 100 °C, as confirmed by X-ray diffraction (XRD) analysis in Supplementary Fig. 2. It is seen that the peaks corresponding to $Sb_2Te_3$ and Te phases in the XRD pattern (Supplementary Fig. 2) become sharper with increasing temperatures, suggesting the suitability as a sintering aid. The absence of peak representing the decomposition of the ChaM in thermogravimetric analysis (TGA) of the $Sb_2Te_3$ ChaM dried at room temperature (Supplementary Fig. 3) implies that it was completely decomposed during drying process[27].

The $Sb_2Te_3$ ChaM can be dispersed in various polar solvents as long as their dielectric constant ($\varepsilon$, F m$^{-1}$) ranges from 10 to 50 for example, dimethyl sulfoxide ($\varepsilon \approx 47$), dimethylformamide

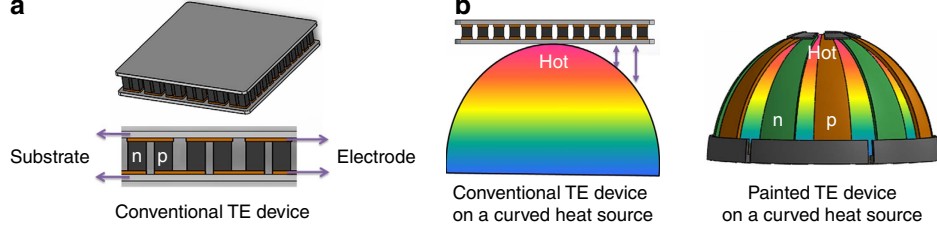

**Figure 1 | Comparison of power generation between the conventional planar-structured TE generator and the painted TE generator on a curved heat source.** (**a**) A conventional planar-structured TE device. (**b**) Scheme of power generation of the conventional TE generator and the painted TE generator on a curved heat source.

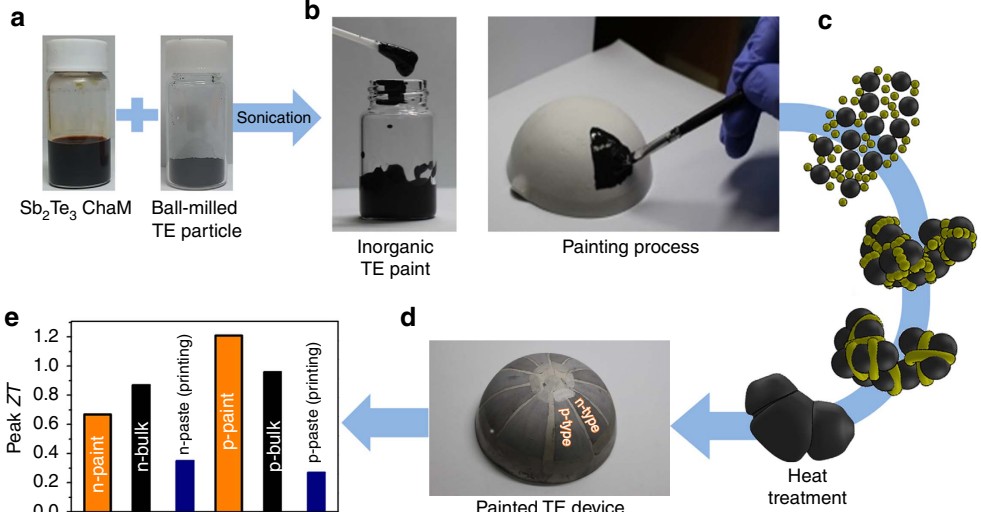

**Figure 2 | Schematic illustrating for the fabrication of painted TE devices with Bi$_2$Te$_3$-based inorganic TE paints.** (**a**) Photographs of the Sb$_2$Te$_3$ ChaM solution and the ball-milled TE particle. (**b**) Photographs for the fabricated TE paint and the painting process on an alumina hemisphere. (**c**) Scheme for the sintering of ball-milled particles assisted by the Sb$_2$Te$_3$ ChaM. The molecular Sb$_2$Te$_3$ ChaM ions act as a sintering aid to fill up the void space among ball-milled particles and promote the grain growth and densification. The yellow and black particles indicate the ChaM molecules and ball-milled particles, respectively. (**d**) Photograph of the fabricated hemispherical TE device. (**e**) Comparison of the peak ZT values among the painted n- and p-type materials in the current study, the typical bulk Bi$_2$Te$_3$-based materials (denoted as n-bulk and p-bulk) and the printed materials reported in the literature (denoted as n-paste and p-paste)[2,23].

($\varepsilon \approx 36$), ethylenediamine ($\varepsilon \approx 13$) and viscous polar solvents of ethylene glycol ($\varepsilon \approx 37$) and glycerol ($\varepsilon \approx 43$; Supplementary Fig. 4). This provides a room for tuning the dielectric constant, solvent viscosity and evaporation temperature of the TE paints. We dispersed the Sb$_2$Te$_3$ ChaM (20 wt% of TE particles) in a mixed viscous solvent of glycerol and ethylene glycol containing n-type Bi$_{2.0}$Te$_{2.7}$Se$_{0.3}$ (BTS) or p-type Bi$_{0.4}$Sb$_{1.6}$Te$_{3.0}$ (BST) TE microparticles (Fig. 2a). The viscosity and evaporation temperature for the TE paints were both adjusted by controlling the ratio of glycerol (viscosity at room temperature $\approx 934$ mP s, boiling point $\approx 290\,^\circ$C) to ethylene glycol (viscosity at room temperature $\approx 62$ mP s, boiling point $\approx 197\,^\circ$C). We found that the suspension was stable against phase separation and precipitation for more than a week (Supplementary Fig. 5).

**Sb$_2$Te$_3$ ChaM as a sintering aid**. To fully understand the sintering behaviour of the TE paints, both n- and p-type paints, repeatedly painted and dried, were sintered at various temperatures $>350\,^\circ$C, with all producing mechanically robust TE samples several hundred micrometres in thickness. Figure 3a–d compare the microstructure of n-type BTS and p-type BST TE materials sintered at $450\,^\circ$C with and without the Sb$_2$Te$_3$ ChaM. It is noted that a suspension of TE particles without the ChaM painted and sintered under the same conditions resulted in at most 60–70 % of the density achieved with the ChaM (Fig. 3e), regardless of the sintering temperature. As shown in Fig. 3e, the presence of the ChaM effectively increases the initial density of the TE materials by filling up pores, promoting the grain growth and densification of the ensemble of particles. As a whole, the evidences demonstrate the effectiveness of the Sb$_2$Te$_3$ ChaM as a sintering aid.

While the density of sintered materials kept increasing with temperature, asymptotically approaching 3.9 g cm$^{-3}$ (n-type) and 3.6 g cm$^{-3}$ (p-type) above $400\,^\circ$C (Fig. 3e and Supplementary Fig. 6); however, a TGA profile in Supplementary Fig. 7 revealed that a weight loss occurs above $450\,^\circ$C due to the evaporation of liquid Te, which is known to result in a slight

degradation of properties via the formation of the Te vacancy defect[43]. Therefore, the optimum sintering temperature for the current study was taken at $450\,^\circ$C.

As manifested from the microstructures (Fig. 3a–d), the grain morphology clearly dictates that the grain growth took place in a layer-by-layer mode, which requires two-dimensional nucleation event from a liquid medium as a prerequisite[44]. The scanning electron microscope (SEM) image of the fractured surface (Supplementary Fig. 8) shows the stereotypical microstructure formed by a nucleation and lateral growth[44]. This implies that the added sintering aid formed a liquid phase at the sintering temperature, which provides a diffusion path for grain growth. As evidenced by the differential scanning calorimetry curves of n-type and p-type paints (Supplementary Fig. 7), the Te phase formed from the Sb$_2$Te$_3$ ChaM sintering aid is melted at $\sim 420\,^\circ$C, lower than the sintering temperature of $450\,^\circ$C. It means that the liquefied Te can contribute to the liquid-phase sintering on heat treatment. A possible contribution from the viscous flow mechanism during the initial stage of the liquid-phase sintering was ruled out based on an analysis on a time-dependent shrinkage measurement as shown in Supplementary Fig. 9, where the time exponent of 0.08 is determined to be much smaller than the theoretically expected one. It is noted that the viscous flow mechanism during liquid-phase sintering is often represented as the following relation[45]: $\Delta l/l \propto t^{1+y}$, where $l$ and $t$ denote a linear dimension of the sample and sintering time, respectively. Here, the exponent $1 + y$ is slightly larger than unity due to increasing driving force with decreasing pore size during the process.

The temperature-dependent XRD patterns (Supplementary Figs 10 and 11) demonstrate that the Sb$_2$Te$_3$ ChaM was completely integrated into the host phase, suggesting that the Sb$_2$Te$_3$ ChaM should be compositionally compatible with the growth unit of the host phases. It is more pronounced in n-type materials. The XRD patterns (Supplementary Fig. 10) shows the peak shift to lower angle with increasing the sintering temperature, signifying the increase of Te stoichiometric ratio in a Bi$_2$(Te,Se)$_3$ phase due to the integration of the Sb$_2$Te$_3$ ChaM into

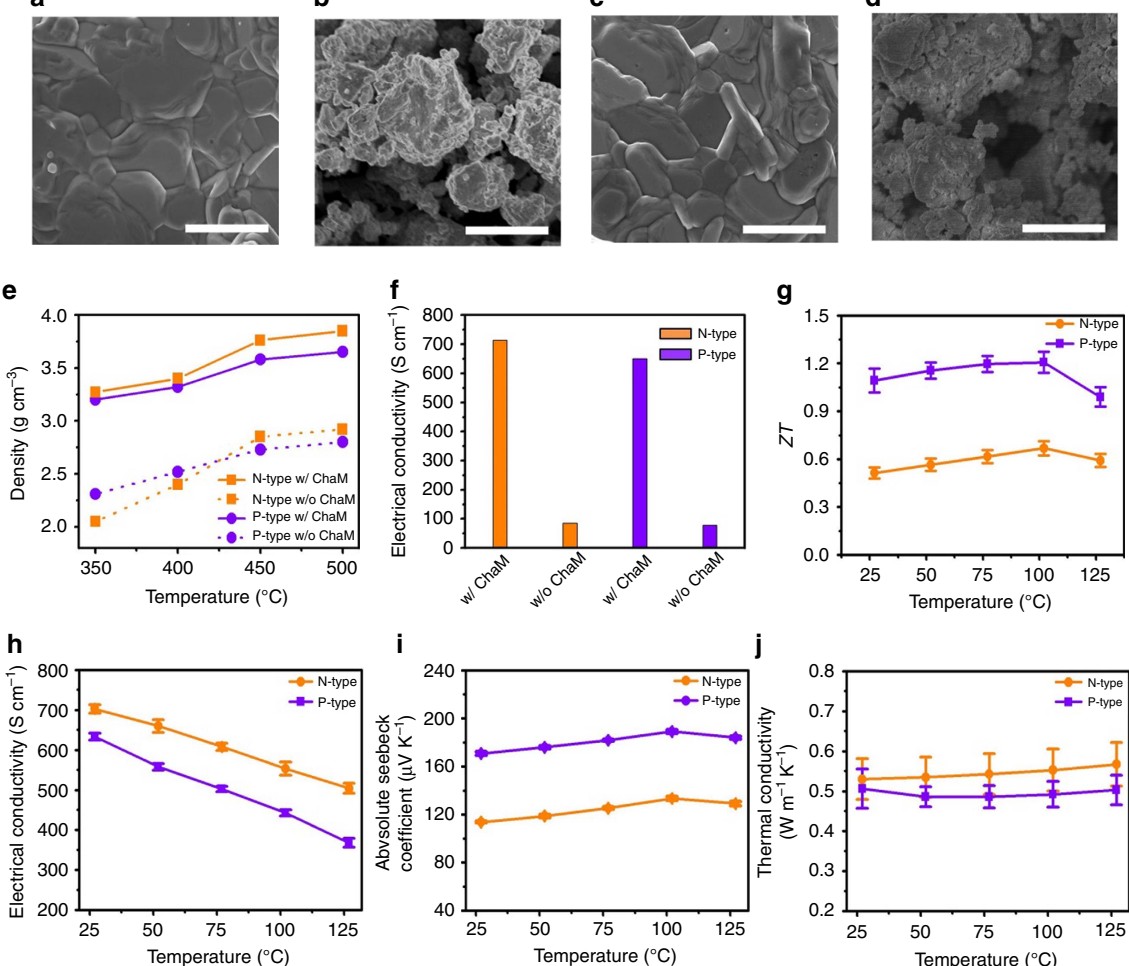

**Figure 3 | Characterization of the painted TE materials.** SEM images of the painted n-type materials (**a**) with and (**b**) without the ChaM, and the p-type materials (**c**) with and (**d**) without the ChaM. All scale bars are 5 μm (**a–d**). Comparison of (**e**) density and (**f**) electrical conductivity between the materials with and without the ChaM. Temperature-dependent TE properties of n- and p-type painted materials: (**g**) ZT value (**h**) electrical conductivity (**i**) absolute Seebeck coefficient and (**j**) thermal conductivity. Error bars represent the s.e. of the mean values of ZT, electrical conductivity, Seebeck coefficient and thermal conductivity obtained by the measurement of three different samples.

the host phase. The significance of the improved sinterability is best-reflected in the electrical charge transport property, which is an order of magnitude higher electrical conductivity at $650-750 \, \mathrm{S \, cm^{-1}}$ than those of the materials without the ChaM (Fig. 3f).

**TE properties of the painted materials.** Excellent TE properties are achieved in both the n- and p-type painted TE samples over the temperature range from 25 °C to 125 °C. The room-temperature ZT values of the n- and p-type samples marked 0.51 and 0.97, respectively (Fig. 3g), where the maximum values reached 0.67 for the n-type sample and 1.21 for the p-type sample at 100 °C (Fig. 3g). Note that these maxima are higher than those obtained with typical $\mathrm{Bi_2Te_3}$-based bulk ingots $(ZT \approx 0.8-1.0)^{[1-3]}$ and are close to the recently reported nanostructured TE materials $(ZT \approx 1.1-1.9)^{[46-48]}$. Furthermore, these values are highest among the reported TE materials based on TE inks or pastes, and 3–4 times greater than anything that has been previously reported for printed TE materials (Fig. 2e)[23].

These promising ZT values of the painted samples originate in high electrical conductivities and ultra-low thermal conductivities. The electrical conductivities of the n- and p-type

samples (Fig. 3h) are $650-750 \, \mathrm{S \, cm^{-1}}$ at room temperature, decreasing with increasing temperature. These high electrical conductivities result from the moderately high carrier mobilities of $149 \, \mathrm{cm^2 \, V^{-1} \, s^{-1}}$ for the n-type and $141 \, \mathrm{cm^2 \, V^{-1} \, s^{-1}}$ for the p-type materials. The Seebeck coefficient of the n-type samples (Fig. 3i) is $114 \, \mathrm{\mu V \, °C^{-1}}$ at room temperature with a peak value of $134 \, \mathrm{\mu V \, °C^{-1}}$ at 102 °C, and that of the p-type samples is $170-190 \, \mathrm{\mu V \, °C^{-1}}$ over the entire measurement temperature range (Fig. 3i). These relatively low-Seebeck coefficients are caused by the high carrier concentrations of $3.0 \times 10^{19} \, \mathrm{cm^{-3}}$ for the n-type samples and $2.9 \times 10^{19} \, \mathrm{cm^{-3}}$ for the p-type samples, since the Seebeck coefficient and the carrier concentration are reciprocally proportional[2,3].

The most significant effect of molecular ChaM-assisted sintering is seen in the great reduction in the thermal conductivities of the n- and p-type samples (Fig. 3j), that is, $0.5-0.6 \, \mathrm{W \, m^{-1} \, K^{-1}}$ in comparison with the $1.5-2.5 \, \mathrm{W \, m^{-1} \, K^{-1}}$ of bulk $\mathrm{Bi_2Te_3}$-based materials[2]. The calculated lattice thermal conductivities were as low as $0.19 \, \mathrm{W \, m^{-1} \, K^{-1}}$ for n-type and $0.20 \, \mathrm{W \, m^{-1} \, K^{-1}}$ for p-type painted materials (Supplementary Fig. 12). These values are lower or comparable than the predicted minimum lattice thermal conductivities of $0.31 \, \mathrm{W \, m^{-1} \, K^{-1}}$ in n-type $\mathrm{Bi_2Te_3}$ and $0.20 \, \mathrm{W \, m^{-1} \, K^{-1}}$ and p-type $\mathrm{(Bi,Sb)_2Te_3}$, which is calculated

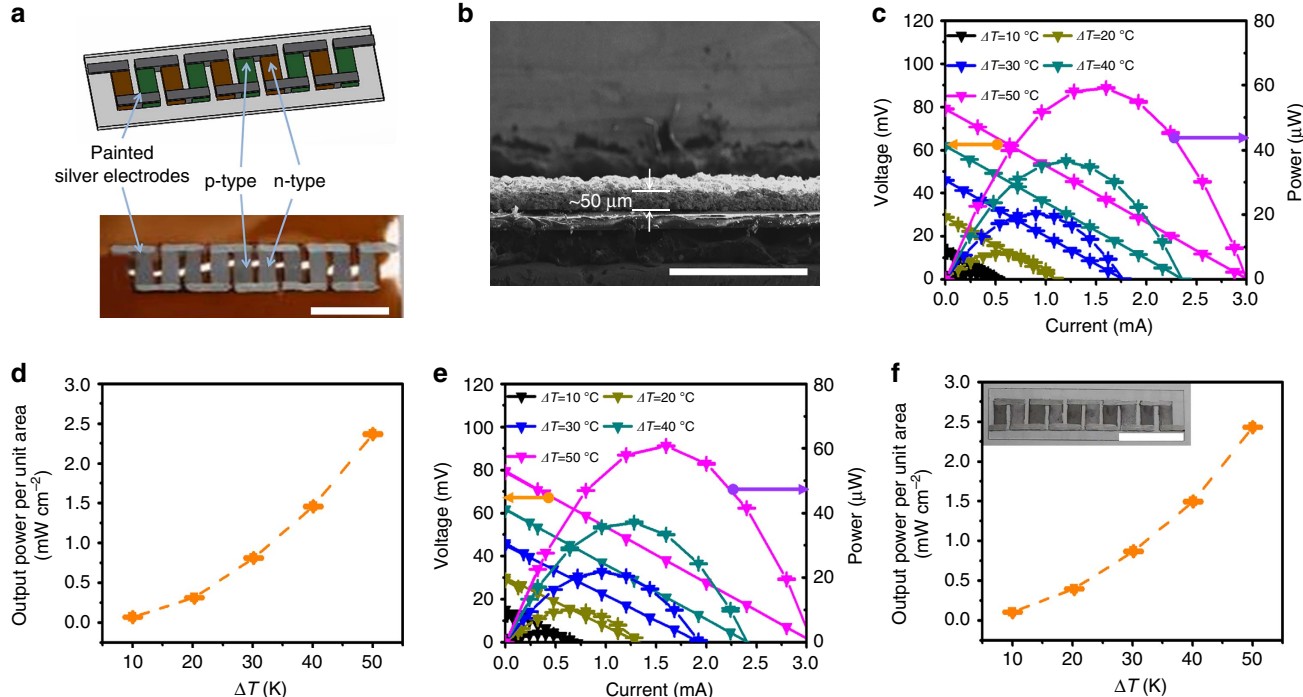

**Figure 4 | Output characteristics of in-plane TE devices painted on flat substrates.** (**a**) Scheme and photograph of an in-plane type TE device composed of painted legs and silver electrodes on a polyimide substrate. (**b**) Cross-sectional SEM image of painted TE layer. Scale bar, 500 μm. (**c**) Output voltage and power, and (**d**) output power density of the painted TE device on a polyimide substrate. (**e**) Output voltage and power, and (**f**) output power density of the painted TE device on a glass substrate (The inset shows a photograph of the actual TE device). Scale bar in (**a**) and the inset of (**f**) is 20 mm. Error bars represent the s.e. of the mean values of output voltage, power and output power density obtained by repeatedly measuring three times. Solid and dashed lines in **c,d,e** and **f** indicate the guide for the measured values.

using the Debye–Callaway model with the assumption of full densities[49]. One possible explanation for the ultra-low lattice thermal conductivity is the porosity of materials. Although the $Sb_2Te_3$ ChaM promotes the sintering of TE particles, their densities are still lower than the bulk values of $6.5–7.5\,g\,cm^{-3}$. The analysis of porosity with $N_2$ adsorption measurement and SEM of these materials (Supplementary Figs 13 and 14) reveal the existence of both nano-scale and micro-scale pores. These multi-scale pores can significantly reduce the thermal conductivity by phonon scattering with a broad range of wavelength at pore sites. To further quantitatively estimate the porosity effect on the thermal transport, the lattice thermal conductivities of painted samples were corrected by using the modified formulation of the effective medium theory suggested by Lee et al.[50]: $\kappa_l = \kappa_h\frac{(2-2\Phi)}{(2+\Phi)}$ , where $\kappa_h$ and $\Phi$ are the lattice thermal conductivity of host materials and the porosity, respectively. The overall porosities of the painted samples were estimated by the direct method of comparing the sample density to the theoretical density of bulk materials with identical compositions. Using the bulk densities of BTS ($7.55\,g\,cm^{-3}$) for the n-type and BST ($6.785\,g\,cm^{-3}$) for the p-type materials, the calculated porosities of the painted samples were 0.47 for the n-type and 0.46 for the p-type samples. The calculated minimum lattice thermal conductivities of the n-type and p-type painted samples are $0.44\,W\,m^{-1}\,K^{-1}$ and $0.47\,W\,m^{-1}\,K^{-1}$, (Supplementary Fig. 12) respectively, which are comparable to those of typical nanostructured bulk materials prepared from ball-milled $Bi_2(Te,Se)_3$ and $(Bi,Sb)_2Te_3$.

Generally, the porosity of solid materials strongly affects the charge carrier transport due to scattering of carriers at the pore sites[51]. A charge carrier passing near a pore is scattered due to the potential perturbation[50], degrading the carrier mobility and eventually the electrical conductivity. The carrier-scattering effect

on mobility can be qualitatively described by the Matthiessen's rule[52]

$$\frac{1}{\mu_{tot}} = \frac{1}{\mu_{bulk}} + \frac{1}{\mu_{impurity}} + \frac{1}{\mu_{boundary}} + \frac{1}{\mu_{pore}} \quad (1)$$

Accordingly, the total scattering is the sum of the contribution of different carrier scattering mechanism. For example, $\mu_{bulk}$ is the mobility induced solely by the carrier scattering with acoustic phonons. In the painted materials, considering no additional impurity element except Bi, Sb and Te, $\mu_{boundary}$ and $\mu_{pore}$ should be the critical factors to determine the overall mobility. Lee et al.[50] suggested that the porosity effect on electrical properties become weaker for larger grains. Since the material with larger grains necessarily has larger pores with the lower number density under the same porosity, the scattering rate is reduced and mobility is enhanced for larger grain sizes. The fact that the grain size is in the range of several micrometres (Fig. 3a,c) and the pores are mainly macro-scale in the painted materials (<3% of micro-pores in volume) suggests that the moderately high mobility is attributed to the lower number density of the pore.

Another important factor to determine the electrical conductivity is the carrier concentration. To overcome the lower mobility of the painted samples than those of bulk, we chose the composition of BST (p-type) and BTS (n-type) for host matrix materials. The materials with such compositions are known to exhibit high carrier concentration by the formation of $Sb_{Te}$ antisite defect to provide hole in p-type and Se vacancy defect to provide electron in n-type. In fact, the carrier concentrations of the painted samples were two or threefold higher than $1\sim2\,cm^{-3}$ of typically used $Bi_2Te_3$-based materials[2]. Although these high carrier concentrations decreased the Seebeck

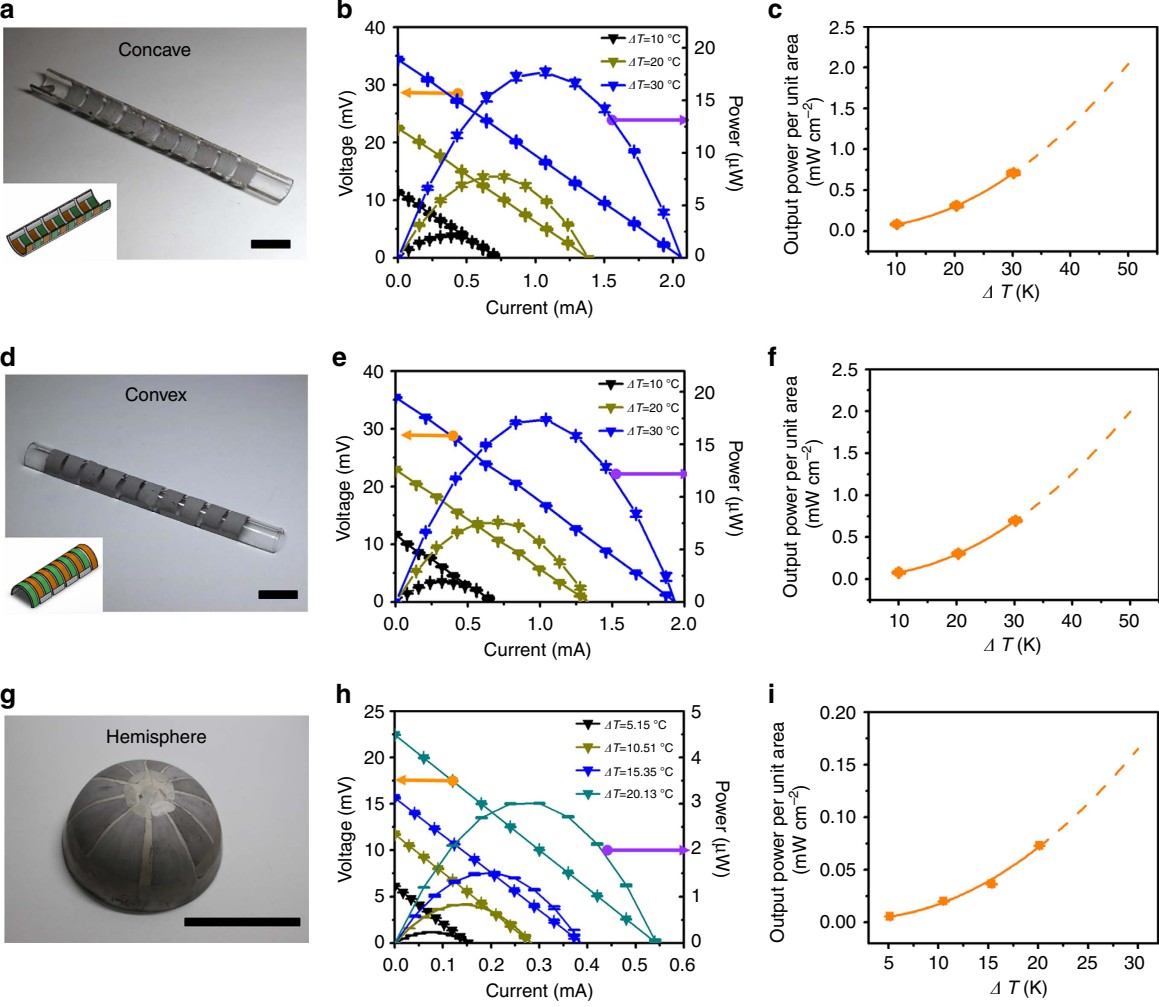

**Figure 5 | Output characteristics of in-plane TE devices painted on curved substrates.** (**a,d,g**) Photographs of the painted TE devices on concave and convex surfaces of a hemi-cylinder and a hemisphere (the insets of **a** and **d** show schemes of the TE devices). Scale bars in (**a**), (**d**) and (**g**) are 10, 10 and 50 mm, respectively. (**b,e,h**) Linear and curved lines indicate output voltages and powers, and (**c,f,i**) output power densities of the painted TE devices on concave and convex surfaces of a hemi-cylinder, and a hemisphere. Error bars represent the s.e. of the mean values of output voltage, power and output power density obtained by repeatedly measuring three times. Solid and dashed lines in **b,c,e,f,h** and **i** indicate the guide for the measured values and the predicted properties via extrapolations.

coefficients, the electrical conductivities were significantly increased up to 650–750 S cm$^{-1}$ at room temperature, close to bulk values. Consequently, in spite of high porosity, the high carrier concentration, the low number density of pores and bulk-scale grains can result in the high electrical conductivity of the painted materials.

**All-painted TE generating devices on flat surfaces.** The outstanding TE properties and painting processability of these TE paints make it possible to design highly efficient TE generating devices geometrically compatible with heat sources. As the first attempt, n- and p-type TE paints were applied with a brush to a flexible polyimide substrate, and then sintered at 450 °C for 10 min (Fig. 4a). These painted layers formed continuously uniform films with the thickness of about 50 μm (Fig. 4b). It means that the sintering condition is enough for the system to reach the final stage of sintering, where the coarsening process becomes stagnant with a narrow size distribution. Ag paste was painted in a way that the TE device consists of 5 couples of n- and p-type legs with lateral dimensions of 5 mm × 10 mm and an average thickness of ~50 μm (Fig. 4a). The internal resistance

of this device was 25.8 Ω, higher than the expected resistance in reference to the electrical properties, suggesting that the contact resistance between the Ag electrode and the TE leg is considerably high. We measured the contact resistance between the Ag electrode and the painted TE leg by the transmission line method (Supplementary Fig. 15). The measured contact resistance is quite high at $4.8 \times 10^{-2}\,\Omega\,cm^2$, which is three or four orders of magnitude higher than the contact resistance observed in conventional module composed of Bi$_2$Te$_3$-based TE legs[53] and can be responsible for the internal resistance of the painted TE generator. To ensure reliable evaluation of an output power of this device, only the temperature of the hot side was modulated, while the cold side was kept at a constant temperature of 20 ± 0.5 °C (Supplementary Fig. 16).

The TE device painted onto the polyimide substrate achieved an output voltage of 79.4 mV and an output power of 60.8 μW under the temperature difference of 50 °C (Fig. 4c). The output power density reached as high as 2.43 mW cm$^{-2}$ (Fig. 4d), which doubles the best reported value for in-plane type TE devices[33]. The mW-level output power density is highly potential for the wearable TE energy harvester. Another TE device was prepared by painting onto a glass substrate under the same preparation

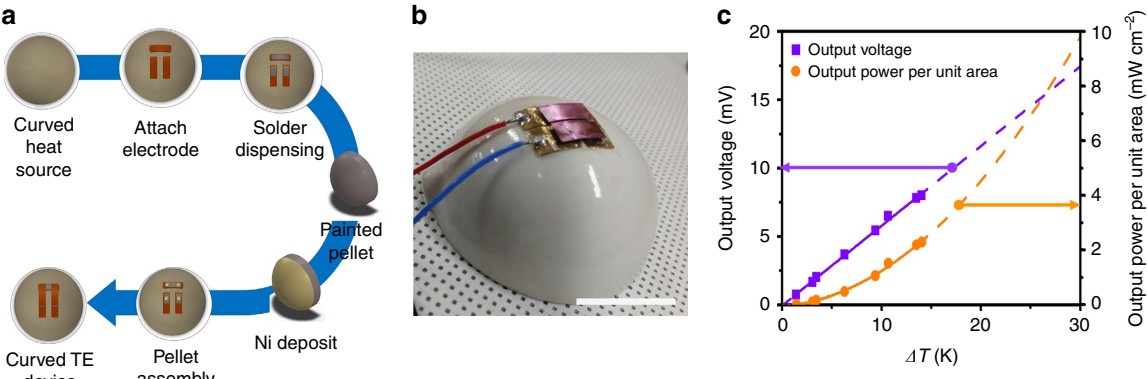

**Figure 6 | Through-plane TE generator on an alumina hemisphere using the moulded pellets prepared from the TE paints.** (**a**) Scheme for the fabrication of the TE generator. (**b**) A photograph of the fabricated TE generator. (**c**) Output voltage and output power density. The scale bar in **b** is 35 mm. Solid and dashed lines in **c** indicate the guide for the measured values and the predicted properties via extrapolations.

conditions. This achieved the internal resistance of 26.3 Ω, the output voltage of 79.4 mV, the output power of 60.7 μW and the output power density of 2.43 mW cm$^{-2}$ (Fig. 4e,f). The fact that these values are almost identical to the TE device painted onto a polyimide substrate suggests that the fabrication process can be consistently applied to a range of different substrates.

**All-painted TE devices on curved surfaces**. The versatility of the TE paints was best-demonstrated by TE devices directly realized onto curved surfaces such as onto the concave and convex surfaces of a glass hemi-cylinder, as depicted in Fig. 5. The resulting device has 5 couples of n- and p-type layers with the dimension of 5 mm × 10 mm × ~50 μm (Fig. 5a,d). The internal resistances of the TE devices on convex and concave surfaces were almost identical at 23–25 Ω, which is consistent with the TE devices painted on flat substrates. Under the temperature difference of 30 °C, these devices produced an output voltage of 34–36 mV and output power of 17–18 μW, leading to a comparable power density (0.70–0.71 mW cm$^{-2}$) to the TE devices painted on flat glass substrates (Fig. 5b,c for the concave device, and Fig. 5e,f for the convex device). The fact that the output power density of the TE devices painted on flat and curved substrates with the same dimension of TE legs merges into the same line (Supplementary Fig. 17) validates the applicability of the TE paints to any shaped surfaces.

To further demonstrate the processability of TE painting onto large-sized curved surfaces with a full coverage, TE device was fabricated on a hemispherical alumina substrate with the diameter of ~70 mm (Fig. 5g). We introduced 5.5 couples of triangular TE layers with 15 mm at the base and ~25 mm in height and obtained the internal resistance of 40.2 Ω, which is expected for the enlarged TE layers (67% higher aspect ratio). To minimize radiation or convection factor from a heat source, the planar heat source was fully covered with a glass fabric and the apex of the hemispherical generator was thermally connected by thermal pads (Supplementary Fig. 18). Exposed to a temperature difference of 20.1 °C, this device produced the output voltage of 22.5 mV, the output power of 3.0 μW and the output power density of 0.073 mW cm$^{-2}$ (Fig. 5h,i), which are significantly lower than those of the other devices. It is understood that this low output power density can originate in the longer TE legs which increase the internal resistance since the output power density is inversely proportional to the leg length under same temperature difference[54]. Assuming the identical dimension of the TE legs to those of other

devices, the plotted output power density approached towards the others (Supplementary Fig. 17). As evidenced by the lower output voltage, small deviation in the graph (Supplementary Fig. 17) can be due to the heat loss from a thermally conducting alumina substrate, which forces the external temperatures to be different from the actual temperature applied to the TE legs.

**Comparison with conventional TE module**. To show how the painted TE generator on a curved surface is effective, we performed the comparative simulation study on the power output of the painted TE generator and the conventional module on a hemispherical curved heat source, based on a three-dimensional TE finite element model (FEM). The heat loss in the FEM was considered by including the convective heat transfer. To simulate the natural convection over all the surfaces that are exposed to air, the convection heat transfer coefficient was 10 W m$^{-2}$ K$^{-1}$ with an ambient temperature of 25 °C (ref. 55). The simulation details are described in the Supplementary Information. The temperatures across the apex and the bottom of an alumina hemisphere were kept at 45 and 25 °C (Supplementary Fig. 19). In the conventional module, since the contact area ($d$) with a hemisphere is small, the temperature distribution in the module is greatly non-uniform (Supplementary Fig. 20), which results in a significantly low output voltage of 13.3 mV for $d = 1$ mm and 4.5 mV for $d = 0.1$ mm. Thus, conventional module generates the output power of 76.9 μW (the output power density of 15 μW cm$^{-2}$) when $d = 1$ mm, and the output power of 8.6 μW (the output power density of 1.7 μW cm$^{-2}$) when $d = 0.1$ mm, which are greatly reduced values compared with the reported values of 4–10 mW cm$^{-2}$ obtained on a flat heat source[5]. On the other hand, the uniform temperature distribution and electrical potential field on the painted generator (Supplementary Fig. 21) result in an order of magnitude higher output power density of 205 μW cm$^{-2}$.

To further validate the practicability of the painting technology against the conventional module, we propose two designs of power generation systems based on the painting technology. First, the TE leg length in the painted TE generator was controllably varied to obtain the higher power output density, since the power output density can be maximized by the optimum TE leg length[5,54]. As shown in Supplementary Fig. 22, with the decrease of the leg length, the resistance linearly decreases and the output power increases as expected. The highest output power per unit area of 4.0 mW cm$^{-2}$ was achieved by the generator with the leg length of 5 mm under the temperature difference of 50 °C.

Furthermore, the predicted power output density based on the fitted function with the data points reached $11.0\,\mathrm{mW\,cm^{-2}}$ in the generator with the leg length of 1.4 mm, which is comparable to $30–50\,\mathrm{mW\,cm^{-2}}$ obtained from the conventional module with TE legs with an identical length on a flat heat source[5].

In addition, we fabricated the through-plane TE generator using the moulded disks prepared from the TE paints. The details of the moulding experiments are described in the Supplementary Information (Supplementary Fig. 23). Two pairs of n-type and p-type moulded disks with the diameter of 4.0 mm and thickness of 1.0 mm were assembled by soldering with a Bi–Sn solder to Cu foil electrodes on an alumina hemisphere (Fig. 6a,b). The internal resistance was as low as $0.014\,\Omega$, comparable to that of the conventional module. Under the temperature difference of 14 °C, this generator produced an output voltage of 8.0 mV, output power of 1.1 mW and output power density of $2.3\,\mathrm{mW\,cm^{-2}}$ (Fig. 6c). Furthermore, the predicted power output density on the fitted function with the data points is as high as $26.3\,\mathrm{mW\,cm^{-2}}$ under the temperature difference of 50 °C, which competes on par with the conventional module[5]. These results clearly demonstrate the practicability of the painting technology in terms of the TE performance as well as the processability.

## Discussion

In summary, this study demonstrates that the painted TE devices exhibiting high output power of $4.0\,\mathrm{mW\,cm^{-2}}$ can be easily prepared on any-shaped surfaces by the molecule-level sintering effect. The most pronounced effect of the molecular sintering aid was shown to enhance the sinterability of TE particles, leading to the particularly high $ZT$ values of 0.67 and 1.21 for n- and p-type painted materials, respectively. It was shown that the versatility of the TE paints roots in the unlimited degrees of freedom for dimension and shape engineering in 2D and 3D structures without killing TE properties, shedding a light on the TE device community. Furthermore, this approach has a potential for cost-effective manufacturing of well-designed TE devices depending on heat sources. We strongly believe that the currently developed technology can be easily transferred to other communities such as 3D printed electronics and painted electronic artworks.

## Methods

**Materials.** Pure elemental granules of Bi, Sb, Te and Se ($>99.999\%$) were purchased from 5N Plus. Ethanethiol ($>97\%$), ethylenediamine (En, $>99.5\%$), acetonitrile ($>99.8\%$), glycerol ($>99.8\%$), ethylene glycol ($>99.8\%$), dimethyl sulfoxide ($>99.9\%$) and dimethylformamide ($>99.8\%$) were purchased from Aldrich Chemical Co. All elements and chemicals were used without further purification.

**Synthesis of Bi₂Te₃-based inorganic TE paints.** The synthesis of the Sb₂Te₃ ChaM solution was performed in a N₂-filled glove box. To synthesize the Sb₂Te₃ ChaM solution, elemental Sb (0.32 g) and Te (0.68 g) powder with stoichiometric ratio of Sb₂Te₄ were dissolved in mixed co-solvent including 2 ml of ethanethiol (97%) and 8 ml of ethylenediamine ($>99.5\%$). After stirring for over 6 h, all elemental Sb and Te were fully dissolved in solvent and the resulting solution showed a dark purple colour. Elemental analysis of the Sb₂Te₃ ChaM using inductively coupled plasma optical emission spectrometry (ICP-OES) revealed an overall ratio of Sb to Te of 2/1, which is identical to the initial elemental ratio. And 40 ml of acetonitrile was added into the Sb₂Te₃ ChaM solution, followed by the centrifuge at 7,500 r.p.m. for 10 min. After the centrifuge, the precipitated Sb₂Te₃ ChaM was added into mixed solvents including 3.6 g of glycerol and 0.4 g of ethylene glycol and it was sonicated for 10 min, which produced a dark-brown coloured solution. Bi₂Te₃-based TE powders were prepared by a mechanical alloying process. Typically, finely ground Bi, Sb, Te and Se powder was weighed according to the stoichiometric ratios of BTS for a n-type paint and BST for a p-type paint under N₂ atmosphere, and they were ball-milled with stainless steel balls including two balls with 0.5 inch in diameter and four balls with 0.25-inch in diameter (SPEX, 8000M Mixer/Mill) for 4 h. The formation of BTS and BST alloys were confirmed by the XRD analysis (Supplementary Fig. 24). We carried out a sieving process at 45 μm to remove some large-sized BTS or BST particles. 4.0 g of

TE powders were added into the Sb₂Te₃ ChaM solution, followed by the sonication for 1 h, which produced black-coloured viscous TE paints. The viscosity and evaporation temperatures were adjusted by controllably varying the amount of glycerol and ethylene glycol.

**TE painting process.** All procedures were performed in a N₂-filled glove box. The synthesized n- and p-type TE paints were painted on glass, aluminium, polyimide and alumina substrates with a flat painting brush with the width of 5 mm. Painted layers on a substrates were sequentially dried on a hot-plates at 90 °C for 30 min, 120 °C for 30 min and 160 °C for 30 min and then they were annealed at desired temperatures (350–450 °C) for 10–30 min. To obtain thick painted layers with several hundreds of micrometre in thickness, painting and drying processes were repeated by several times, followed by annealing.

**Fabrication of painted TE devices.** In-plane type TE devices were fabricated by painting five couples of n- and p-type TE paints with the size of 5 mm × 10 mm on various substrates such as flat glass and polyimide, and curved hemi-cylindrical convex and concave glasses, and alumina hemisphere. Glass and alumina substrates were hydrophilized by a ultraviolet plasma treatment for 1 h before painting. Silver paste was painted electrically in series and thermally in parallel to interconnect n- and p-type TE legs (Fig. 4a). All of the procedures were performed in nitrogen-filled glove box.

**Fabrication of through-plane TE devices.** Through-plane type TE devices were fabricated by using the moulded disks prepared from the TE paints. The details of the moulding are described in Supplementary Fig. 23. Two pairs of n-type and p-type disks with the diameter of 4.0 mm and thickness of 1.0 mm were soldered using a Bi–Sn solder to the pre-patterned Cu foil electrodes on an alumina hemisphere (Fig. 6a). The top sides of TE disks were electrically interconnected with Cu foil electrodes by soldering (Fig. 6a), which produced the through-plane TE generator on a hemisphere.

**TE properties measurement.** TE properties measurement were performed on the samples prepared by repeated painting and drying of n- and p-type paints on aluminium plates, and subsequent annealing at 450 °C. The final samples were $\sim 500\,\mu\mathrm{m}$ in thickness. Furthermore, we characterized the TE properties of more than three sets of n-type and p-type painted samples and added s.e. to each data points. The uncertainties of the electrical conductivity, Seebeck coefficients and thermal conductivity were 1.5, 1.0 and 5.9%, respectively, which demonstrates the reproducibility of the painting technology. To determine electrical conductivities at temperatures ranging from 27 to 127 °C, the sheet resistance of the samples was measured by a four-point Van der Pauw method (Keithley 2,400 multimeter controlled Lab trace 2.0 software, Keithley Instrument, Inc.) on a hot chuck plate. The four corners of the samples were contacted by sharp tips controlled by manipulators. The electrical conductivities were estimated with the thickness of the samples. To obtain the temperature-dependent Seebeck coefficients, the open circuit voltage and the temperature gradient were measured by two T-type thermocouples using a Keithley 2,400 source-meter and a Keithley 2,000 multimeter. The measuring set-up lied on a hot-plate and the measuring temperatures were controlled by heating a hot-plate. To apply the temperature differences, applied powers of commercial TE modules contacted with the samples was adjusted. Typically, six data points were obtained with the applied temperature differences and resulting voltages at two points contacted by thermocouples across the sample ranging from $\pm 1\,°\mathrm{C}$ to $5\,°\mathrm{C}$. The Seebeck coefficient was calculated based on the slope of the voltage versus the temperature-difference curves. This set-up was confirmed by measuring the Seebeck coefficient of n-type Bi₂Te₃ and p-type BiSbTe ingot samples, and the accuracy was within $\pm 3\%$. We extracted the thermal conductivity by using the equation $\kappa = \rho C_P D$, where $\rho$ is the density, $C_P$ is the specific heat capacity and $D$ is the thermal diffusivity. The densities were measured by commercial equipment (BELPycno, microtracBEL). We calculated the specific heat capacities by assuming the law of mixture and by using the value of Bi₀.₄Sb₁.₆Te₄, Sb₂Te₃, Bi₂Te₃ and Bi₂Se₃ (refs 56–58). Thermal diffusivities were measured in a temperature range from 27 to 127 °C by using laser flash analysis (LFA 457, Netzsch). Carrier concentrations and mobilities were measured by a Hall measurement system (BIO-PAD, HL5500PC) at room temperature.

**Simulation study on the power output.** We developed a three-dimensional TE FEM using a commercial software package (COMSOL) for TE generators. The model calculates the temperature distribution and generated power of the TE generators integrated with a heated hemispherical alumina substrate. Similar to the curved structures exposed to sun light, we assumed that the substrate is subject to uniform heat flux, $1.5\,\mathrm{kW\,m^{-2}}$, together with convective heat transfer (Supplementary Fig. 19). To simulate natural convection of air, the convective heat transfer coefficient ($h$) was assumed as $10\,\mathrm{W\,m^{-2}\,K^{-1}}$ against an ambient air at 25 °C. The temperature across the substrate bottom surface was kept at 25 °C. The alumina substrate has a thickness of 2 mm, radius of 38 mm and thermal conductivity of $30\,\mathrm{W\,m^{-1}\,K^{-1}}$. Supplementary Fig. 19b,c show the calculated temperature distribution along an arc A–B where the temperature gradually

decreases from 45 °C to 25 °C. The calculated temperature distribution in Supplementary Fig. 19c was an input to the FEM for the projected TE generator as shown in Supplementary Fig. 19a. To simplify the FEM, the painted TE generator fabricated on the hemispherical substrate was projected on a flat plane. The TE generator consists of one pair of p- and n- types of TE layers and conductive paste layers where the thickness was assumed as 50 μm. Each triangular TE layer has a width of 20 mm and a height of 60 mm such that the substrate has sufficient area for 5.5 couples of TE layers. The conductive paste was assumed to have the thermal conductivity of $9\,W\,m^{-1}\cdot K^{-1}$ and the electrical conductivity of $103\,S\,cm^{-1}$. On the basis of the geometry and the material properties, the electrical resistance was estimated as $3.5\,\Omega$ per a pair of the TE layers.

Commonly, a TE module includes p- and n- types of Bi–Te alloys that are sandwiched by flat alumina substrates. On the basis of the survey of the material properties and the geometry for commercial TE modules, we defined a standard TE module with a substrate area of 40 mm × 40 mm and ∼ 100 pairs of Bi–Te materials. The considered module has the thermal conductance of $0.65\,W\,K^{-1}$, the electrical resistance of $2.3\,\Omega$ and the Seebeck coefficient of $52.8\,mV\,K^{-1}$. For the simple modelling, the FEM for the conventional module includes only one leg of TE material, which has identical physical properties to the properties described above.

**Materials characterization.** *X-ray photoelectron spectroscopy*. The spectrum of the $Sb_2Te_3$ ChaM prepared by vacuum drying at room temperature for a day was obtained using a ThermoFisher K-alpha with a Mg $K_\alpha$ X-ray monochromatic source.

*X-ray diffraction*. XRD patterns were collected by using X'pert Pro, PANalytical with a Cu Kα C-ray source, which has a characteristic wavelength of 1.5418, operating at 40 KV and 30 mA equipped with an X'Celerator detector.

*Differential scanning calorimetry and thermogravimetric analysis*. Differential scanning calorimetry and TGA was simultaneously performed by using Q200 (TA instrument) with a heating rate of 10 °C per min$^{-1}$ under nitrogen flow rate of 100 ml min$^{-1}$.

*Ultraviolet–visible absorption spectroscopy*. The absorption spectra were collected using a Cary 5000 (Agilent) spectrophotometer.

*Scanning electronic microscopy*. The microstructure was characterized by using a field effect SEM (Nova-NanoSEM230, FEI and S-4800 Hitach high-Technologies) operated at 10 KV.

*Elemental mapping*. The elemental maps of vacuum dried samples were characterized with energy-dispersive X-ray spectroscopy by using a Nova-NanoSEM230.

*Brunauer–Emmett–Teller analysis*. Nitrogen adsorption-desorption isotherms of the painted samples were characterized using a physisorption analyzer (ASAP2420, Micromeritics Instruments) The porosities of micro-pores in the painted materials were calculated with the micro-pore volumes obtained by the Brunauer–Emmett–Teller measurement, which were $0.00261\,cm^3\,g^{-1}$ for the n-type sample and $0.00319\,cm^3\,g^{-1}$ for the p-type sample. Also, the average pore sizes were 5.85 nm for the n-type and 5.70 nm for the p-type samples. According to these data and the densities of the samples, the estimated portions of micro-pores in the entire porosity were 2.1% for the n-type sample and 2.5% for the p-type sample, respectively.

**Measurement of TE power generation.** Performance of TE power generator was investigated by measuring the I–V curve and the output power density under temperature differences across the devices was measured using a home-built set-up (Supplementary Fig. 16). To produce a reliable temperature difference ranging from 5 °C to 50 °C across the TE devices, the hot-side temperature was raised using a flat band heater, powered by a voltage converter. The cold side temperature was maintained at 20 ± 0.5 °C using a commercial TE Peltier cooler. The temperature differences were measured by two T-type thermocouples that were in contact with hot and cold sides, by using Keithley 2,000 multimeter. Two Ag electrodes in prepared TE generators were connected to Keithley 2,400 source-meter and the I–V characteristics were measured by using Lab trace 2.0 software (Keithley Instrument, Inc) under desired temperature differences. We also repeatedly measured the output power characteristics of the painted generators by more than three times and add error bars in each data points in the revised manuscript. The output power density (output power per unit area) was calculated with total cross-sectional areas of TE layers.

For the measurement of the in-plane painted hemispherical generator, the planar heat source was fully covered with a glass fabric and the apex of the hemispherical generator was thermally connected by thermal pads to minimize the radiation and convection effects. The power output of the hemispherical generator was comparatively measured with and without a glass fabric covering.

For the measurement of the through-plane hemispherical generator, the hemisphere was heated by a band heater, powered by a voltage converter. For the cold side, the top Cu electrodes were connected to a commercial TE Peltier cooler by thermal pads. For the measurement of the internal resistance, the device was connected to an ammeter in series and to a voltmeter in parallel. The temperature differences were measured by two T-type thermocouples that were in contact with hot and cold sides. The maximum power output ($P$) was calculated by the equation of $P=\frac{V^2}{4R}$, where $V$ is the output voltage and $R$ is the internal resistance, respectively.

**Data availability.** The data that support the findings of this study are available from the corresponding author on request.

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

## Acknowledgements

We acknowledge the financial support by the R&D Convergence Program of NST (National Research Council of Science & Technology) of Republic of Korea, the Center for Advanced Meta-Materials (CAMM) funded by the Ministry of Science, ICT and Future Planning as Global Frontier Project (NRF-2014M3A6B3063704) in Republic of Korea, and Basic Science Research Program through the National Research Foundation of Korea (NRF) funded by the Ministry of Education (2015R1C1A1A01053599).

## Author contributions

S.H.P., S.J., H.W.B., F.K., J.-S.K., D.-B.H., K.J.C., W.J. and J.S.S. designed the experiments, analysed the data and wrote the paper. S.H.P., D.H.G., S.L. and J.S.S. carried out the synthesis and basic characterization of the materials. J.E.L., Y.H. and W.J. performed the measurement of TE properties. S.H.P. and J.S.S. carried out the preparation and measurement of TE devices. All authors discussed the results and commented on the manuscript.

## Additional information

**Competing financial interests:** The authors declare no competing financial interests.

