## [Peer Review File · Nature Communications]

Reviewer #1 (Remarks to the Author)

The comment of this manuscript is as following:

The authors prepare Bi₂Te₃-based inorganic paints using the molecular Sb₂Te₃ chalcogenidometalate as a sintering aid for thermoelectric particles, achieving the peak ZT values of 0.69 for n-type and 1.15 for p-type painted materials that are comparable to the bulk values and three times higher than the best values among the printed TE materials reported in the literature. In addition, the TE generators through the painting TE paints are very effective in heat energy collection from any heat sources with exceedingly high output power density of 2.4 mW/cm², which is the best value among the reported in-plane type TE generators. I think it's a quite good job and recommend it to be published in Nature communications.

Reviewer #2 (Remarks to the Author)

Review comments:

This work improved the material more than a factor of two fold in terms of ZT value as far as the experimental method is appropriate. A criticism for this article, however, may be raised about the performance of the printed modules. The power output per unit area is significantly smaller than (<1/10th of) off-the-shelf conventional modules (30 - 50 mW/cm²) despite with the similar ZT value. Considering a thermal conducting adapter between a wavier surface to a flat conventional thermoelectric module, still one can obtain the power output more than 2.5 mW/cm² on this report. The adapter can be made with a solid heat conduction, a single phase convection, or a two-phase heat transfer like heat pipes. In this point of view, the painting method does not work. Authors need to provide a design supposed to obtain the power output comparable to the conventional modules even on the curved surface.

1. Summary of the key results

Authors provide novel idea of creating a paintable thermoelectric material with lower temperature process and obtained a reasonably good ZT value in comparison with bulk materials.

2. Originality and interest: if not novel, please give references

Originality is fairly good. However, there is a big concern for the interest. See the above comment.

3. Data & methodology: validity of approach, quality of data, quality of presentation

Data and Methodology both are very good and reliable.

4. Appropriate use of statistics and treatment of uncertainties

In treatment of uncertainties are missed. It is no information provided about the repeatability. There is no statistics of the result data found, even repeatability is not described.

5. Conclusions: robustness, validity, reliability

In general, the conclusion of the work is good and the method describe seems to be reliable.

6. Suggested improvements: experiments, data for possible revision

Authors are strongly recommended to provide a design that could have a comparable performance of conventional off-the-shelf flat thermoelectric generators and with the ability of match any curved surfaces. Presented performance does not make sense otherwise.

7. References: appropriate credit to previous work?

This is good.

8. Clarity and context: lucidity of abstract/summary, appropriateness of abstract, introduction and conclusions

Description is fair enough.

Reviewer #3 (Remarks to the Author)

The paper presents shape-engineerable thermoelectric modules which can minimize heat loss caused by the incompatible contact between the heat sources and the thermoelectric modules. The thought is very important for practical applications. However, what distinguish the paper are the scientific spots and the proof of arguments. Here are a few observations.

1. The authors argue that the painted n-type and for p-type legs have high ZT values of 0.69 and 1.15, respectively. These results seem to be sound. However, the major gain of the high ZT values is from the ultralow thermal conductivity (only 0.5–0.6 W m⁻¹ K⁻¹). One can separate the contribution of lattice thermal conductivity which is only 0.1–0.2 W m⁻¹ K⁻¹, apart from the contribution of electronic component. This ultralow lattice thermal conductivity is not convincing because the theoretical minimum thermal conductivity of binary Bi₂Te₃ is about 0.3 W m⁻¹ K⁻¹.
2. The explanation of the ultralow lattice thermal conductivities provided is not reasonable. From the microstructure of sintered materials as shown in Figure 2, the well-sintered grains are in micrometre size. Therefore, the major phonon scattering of the materials should be phonon-phonon interaction rather than boundary scattering in the materials as given in the manuscript. In addition, there is no experimental evidence of the existence of negative curvature boundaries. An additional HRTEM experiment is necessary to observe the negative curvature boundaries.
3. One of the main achievements of this work is using molecular Sb₂Te₃ chalcogenidometalate as a sintering aid to improve the density of the painted thick films hence performance. The most interesting work one expects to see is the sintering mechanism during the aided sintering process. This work is extremely important from the scientific point of view, but is missing in the manuscript.
4. The authors reported that the painted films on the hemispherical alumina substrate show unexpectedly low performance, which has been attributed to the measurement artifact. However, this explanation is a sort of debatable. The heat loss from the highly thermally conducting alumina substrate should be taken into consideration.
5. The large internal resistance of this device as high as 25.8 ohm suggests that the contact resistance between the electrodes and TE legs might be considerably large, according to the high electrical conductivity of the legs. However, no further analysis was provided at this point. While the paper contains interesting technique for the practice applications, I do not find in it the kind of scientific interest one expects to see in a Nature Communications paper.

Response to the reviewers' comments

The followings are the responses to the reviewers' comments for the manuscript "High performance shape-engineerable thermoelectric painting."

▪ Reviewer #1

General comment: The authors prepare Bi₂Te₃-based inorganic paints using the molecular Sb₂Te₃ chalcogenidometalate as a sintering aid for thermoelectric particles, achieving the peak ZT values of 0.69 for n-type and 1.15 for p-type painted materials that are comparable to the bulk values and three times higher than the best values among the printed TE materials reported in the literature. In addition, the TE generators through the painting TE paints are very effective in heat energy collection from any heat sources with exceedingly high output power density of 2.4 mW/cm², which is the best value among the reported in-plane type TE generators. I think it's a quite good job and recommend it to be published in Nature communications.

▪ Reviewer #2

General comment: This work improved the material more than a factor of two fold in terms of ZT value as far as the experimental method is appropriate. A criticism for this article, however, may be raised about the performance of the printed modules. The power output per unit area is significantly smaller than (<1/10th of) off-the-shelf conventional modules (30 - 50 mW/cm²) despite with the similar ZT value. Considering a thermal conducting adapter between a wavier surface to a flat conventional thermoelectric module, still one can obtain the power output more than 2.5 mW/cm² on this report. The adapter can be made with a solid heat conduction, a single phase convection, or a two-phase heat transfer like heat pipes. In this point of view, the painting method does not work. Authors need to provide a design supposed to obtain the power output comparable to the conventional modules even on the curved surface.

Response:

We agree that the comparable performance of the painted thermoelectric (TE) generator is essential to demonstrate the practicability of TE painting technology. As the reviewer commented, the painted TE generators in this work did not exhibit the output power density as high as that of the off-the-shelf planar-structured module with TE legs exhibiting similar ZT values. However, **we would like to point out that the output power density depends solely on the lengths and the power factors of TE legs, under the fixed temperature difference**. In general, the output power density (w , mW cm⁻²) is found as:

$$w = I^2 m R = \frac{m \sigma S^2}{(1+m)^2 l} \Delta T^2 \quad (1)$$

, where I is the electric current, R is the internal resistance, σ is the electrical conductivity of the leg, m is the ratio of the internal resistance to the external load resistance, l is the length of the leg, ΔT is the temperature difference across the leg.^{A1} By matching the external load resistance to the internal resistance, this formulation is simplified to:

$$w = \frac{1}{4l} PF \Delta T^2 \quad (2)$$

, where PF is the power factor (σS^2) of the leg. Assuming the thermal equilibrium, the output power per density depends solely on two factors of PF and l .

On the other hand, one should note that the ΔT applied to the legs from a heat source strongly depend on the thermal conductivity (κ), l , and the thermal resistance (ψ) between the leg and a heat source: as shown in the following relation:

$$\frac{\Delta T}{(T_s - T_a)} = \frac{l}{l + \kappa(X + Y)} \quad (3)$$

, where $(T_s - T_a)$ is the overall temperature difference, and X and Y are the factors proportional to ψ

at hot and cold sides. Under the same $(T_s - T_a)$, it is the higher ΔT across the leg that the longer l and lower κ cause^{A1}. Therefore, l dependence on the power output density has the trade-off relation between R and ΔT , which requires the optimum l for to maximise the highest power output density (Fig. A1).

Figure A1 | Power output normalized to the maximum as a function of normalized leg length (x-axis) with the assumption of $T_a/T_s=0.1$ and $Z=1$ ^{A1}.

Under the same ΔT , the power output density of the painted generator should be an order of magnitude lower than that of the conventional module because of longer l and lower PF , as summarized in Table A1. However, in the view of thermal energy transfer, the ΔT across the legs in the painted generator will be much higher than that of the conventional module since κ is 3~4 times lower and l is more than 5 times longer, according to eqn. (3). In addition, the painting technology enables the direct deposition and fabrication right on heat sources, which make it possible to form the fully-integrated thermal contacts with even curved heat sources. Therefore, it can further help to create a higher ΔT than that of the conventional planar module, which can result in the greater performance of the painted generator.

Table A1 | Comparison of the TE properties of legs in the painted generator and the conventional generator at room temperature

	Painted generator	Conventional generator
Length (mm)	10	1~1.4
Electrical conductivity ($S\ m^{-1}$)	64000~72000	90000~120000
Seebeck coefficient ($\mu V\ K^{-1}$)	115~170	180~210
Power factor ($\mu W\ cm^{-1}\ K^{-2}$)	9~19	30~50
Thermal conductivity ($W\ m^{-1}\ K^{-1}$)	0.50~0.60	1.5~2.5

To verify this hypothesis, we conducted the comparative simulation study on the power output of the painted TE generator and the conventional module on a hemispherical curved heat source. We developed a three-dimensional TE finite element model (FEM) using a commercial software package

(COMSOL) for TE generators. The model calculated the temperature distribution and generated power of the TE generators integrated with a heated hemispherical alumina substrate. Similar to the curved structures exposed to sun light, we assumed that the substrate was subject to uniform heat flux, 1.5 kW m^{-2} , together with convective heat transfer (Fig. A2a). To simulate natural convection of air, the convective heat transfer coefficient (h) was assumed as $10 \text{ W m}^{-2} \text{ K}^{-1}$ against an ambient air at $25 \text{ }^\circ\text{C}$. The temperature across the substrate bottom surface was kept at $25 \text{ }^\circ\text{C}$. The alumina substrate has a thickness of 2 mm , radius of 38 mm , and thermal conductivity of $30 \text{ W m}^{-1} \cdot \text{K}^{-1}$. Fig. A2b and A2c show the calculated temperature distribution along an arc A-B where the temperature gradually decreases from $45 \text{ }^\circ\text{C}$ to $25 \text{ }^\circ\text{C}$.

Figure A2 | A finite element model for a hemispherical heated substrate. (a) A meshed substrate that is subject to a uniform heat flux of 1.5 kW/m^2 , and natural convection ($h = 10 \text{ W m}^{-2} \text{ K}^{-1}$). The bottom surface is set at $25 \text{ }^\circ\text{C}$. (b, c) Temperature distribution along an arc A-B.

The calculated temperature distribution in Fig. A2c was an input to the FEM for the projected TE generator as shown in Fig. A2a. To simplify the FEM, the painted TE generator fabricated on the hemispherical substrate was projected on a flat plane. The TE generator consists of one pair of p- and n- types of TE layers and conductive paste layers where the thickness was assumed as 50 μm . Each triangular TE layer has a width of 20 mm and a height of 60 mm such that the substrate has sufficient area for 5.5 couples of TE layers. The conductive paste was assumed to have the thermal conductivity of $9 \text{ W m}^{-1}\cdot\text{K}^{-1}$ and the electrical conductivity of 103 S cm^{-1} . Based on the geometry and the material properties, the electrical resistance was estimated as 3.5Ω per a pair of the TE layers. Fig. A3 shows the calculated electrical potential distribution across a pair of TE layers where the output voltage is 2.7 mV. Thus, 5.5 couples of TE layers produce the output power of $11.3 \mu\text{W}$ and the output power density of $205 \mu\text{W cm}^{-2}$.

Figure A3 | Calculated (a) temperature and (b) electrical potential distribution of a pair of p- and n- types of painted TE generator.

In order to show how the painted TE generator on a curved surface is effective, we also calculated the generating performance of the conventional TE module. Commonly, a TE module includes p- and n- types of Bi-Te alloys that are sandwiched by flat alumina substrates. Based on the survey of the material properties and the geometry for commercial TE modules, we defined a standard TE module with a

substrate area of 40 mm × 40 mm and ~100 pairs of Bi-Te materials. The considered module has the thermal conductance of 0.65 W K⁻¹, the electrical resistance of 2.3 Ω, and the Seebeck coefficient of 52.8 mV/K. For the simple modelling, the FEM for the conventional module includes only one leg of TE material which has identical physical properties to the properties described above.

Fig. A4 shows the calculated temperature and electrical potential distributions of the standard TE module located on the heated hemispherical substrate. Since the TE module is flat, only a small area of the TE module contacts with the curved surface. We assumed that the contact area has a diameter (d) of either 1 mm (Fig. A4a and A4b) or 0.1 mm (Fig. A4c and A4d). In addition, the TE module was assumed to be under natural convection. Since the contact area is small, the temperature distribution in the TE module is highly non-uniform, which resulted in non-uniform electrical potential field. Although the local maximum of the electrical potential is 207.9 mV for d = 1 mm, and 74.5 mV for d = 0.1 mm, the averaged potential difference across the TE leg reduces to 13.3 mV for d = 1 mm and 4.5 mV for d = 0.1 mm. Thus, conventional module generates the output power of 76.9 μW (the output power density of **15 μW cm⁻²**) when d = 1mm, and the output power of 8.6 μW (the output power density of **1.7 μW cm⁻²**) when d = 0.1mm. Consequently, these values obtained in the conventional module are one order of magnitude lower than **205 μW cm⁻²** of the painted generator, which is attributed to uniform temperature distribution across the painted legs. This result suggests that the painting technology can be more effective for the TE power generation than the conventional module within the context of thermal energy transfer.

Figure A4 | Calculated (a,c) temperature and (b,d) electrical potential distribution of a conventional TE module that is contact on a heated hemispherical substrate. The contact area between the TE module and the heated substrate has a diameter of either 1 mm (a, b), or 0.1 mm (c, d).

Table A2 | Comparison of the calculated power output of the painted generator and the conventional generator on a hemispherical curved heat source with the temperatures of 45 °C (hot side) and 25 °C (cold side).

	Painted generator	Conventional generator	
Contact area (mm)	-	0.1	1
Output voltage (mV)	14.9	4.5	13.3
Power output (μW)	11.3	8.6	76.9
Power output per unit area ($\mu\text{W cm}^{-2}$)	205	1.7	15

As the reviewer commented, the use of thermal conducting adapter such as a solid heat conduction, a single phase convection, or a two-phase heat transfer like heat pipes can improve the thermal energy transfer from even curved heat sources to the planar-structured conventional module. To further validate the practicability of the painting technology, we propose two different power generation designs based on the TE painting technology. First, the l in the painted TE generator was controllably varied to obtain the higher power output density, based on eqn. (2). We fabricated four different painted generator with one-pair of n- and p-type legs with the l from 5 mm to 20 mm and compared their resistances and output power densities under ΔT of 50 °C. As shown in Fig. A5, with the decrease of l , the resistance linearly decreases and the output power increases as expected. The highest output power per unit area of **4.0 mW cm⁻²** was achieved by the generator with the l of 5 mm. Furthermore, the predicted power output density based on the fitted function with the data points reached **11.0 mW cm⁻²** in the generator with the l of 1.4 mm.

Figure A5 | Leg length dependence on output characteristics of in-plane painted TE generator with one pair of TE legs. (a) Output voltage and power, and (b) output power density. The inset of (b) indicates the internal resistance.

In addition to the length-controlled painted generator, we fabricated through-plane TE generator using the moulded disks prepared from the TE paints. Two pairs of n-type and p-type disks with the diameter of 4.0 mm and thickness of 1.0 mm were soldered using a Bi-Sn solder to the pre-patterned Cu foil electrodes on an alumina hemisphere (Fig. A6a). The top sides of TE disks were electrically interconnected with Cu foil electrodes by soldering (Fig. A6a), which produced the through-plane TE generator on a hemisphere. The internal resistance was as low as 0.014Ω , comparable value to that of the conventional module. Under ΔT of $14 \text{ }^\circ\text{C}$, this generator produced an output voltage of 8.0 mV , output power of 1.1 mW , and output power density of 2.3 mW cm^{-2} (Fig. A6b). In particular, the predicted power output density on the fitted function with the data points is as high as **26.3 mW cm^{-2}** under ΔT of $50 \text{ }^\circ\text{C}$, which competes on a par with the conventional module. These results clearly demonstrates the practicability of the painting technology in terms of TE performance as well as processability. Furthermore, we believe that the benefit of the current technology such as the design flexibility of TE materials will attract the attentions in the TE community, which would further boost this technology to compete with the conventional module in the future.

Figure A6 | Through-plane TE generator using moulded pellets prepared from the TE paints. (a) Scheme for the fabrication of the TE generator. (b) Output voltage and output power density. The inset of (b) shows the photograph of the TE generator.

We added **the simulation result, the output power characteristics and schemes for the length-dependent in-plane generators and through-plane generators** in Fig S18-S20, Fig. S21, and Fig. 5 and **the following chapter and the related experimental details** were included in the revised manuscript (page 12-13) and Supplementary Information.

“Comparison with conventional TE module. In order to show how the painted TE generator on a curved surface is effective, we conducted the comparative simulation study on the power output of the painted TE generator and the conventional module on a hemispherical curved heat source, based on a three-dimensional TE finite element model (FEM). The simulation details are described in the Supplementary Information. The temperatures across the apex and the bottom of an alumina hemisphere

were kept at 45 °C and 25 °C (Fig. S18). In the conventional module, since the contact area (d) with a hemisphere is small, the temperature distribution in the module is greatly non-uniform (Fig. S19), which results in a significantly low output voltage of 13.3 mV for $d = 1$ mm and 4.5 mV for $d = 0.1$ mm. Thus, conventional module generates the output power of 76.9 μ W (the output power density of 15 μ W cm^{-2}) when $d = 1$ mm, and the output power of 8.6 μ W (the output power density of 1.7 μ W cm^{-2}) when $d = 0.1$ mm, which are greatly reduced values compared with the reported values of 4-10 mW cm^{-2} obtained on a flat heat source⁵³. On the other hand, the uniform temperature distribution and electrical potential field on the painted generator (Fig. S20) result in an order of magnitude higher output power density of 205 μ W cm^{-2} .

To further validate the practicability of the painting technology against the conventional module, we propose two designs of power generation systems based on the painting technology. First, the TE leg length in the painted TE generator was controllably varied to obtain the higher power output density since the power output density can be maximized by the optimum TE leg length^{52,53}. As shown in Fig. S21, with the decrease of the leg length, the resistance linearly decreases and the output power increases as expected. The highest output power per unit area of 4.0 mW cm^{-2} was achieved by the generator with the leg length of 5 mm under the temperature difference of 50 °C. Furthermore, the predicted power output density based on the fitted function with the data points reached 11.0 mW cm^{-2} in the generator with the leg length of 1.4 mm, which is comparable to 30-50 mW cm^{-2} obtained from the conventional module with TE legs with an identical length on a flat heat source⁵³.

In addition, we fabricated the through-plane TE generator using the moulded disks prepared from the TE paints. The details of the moulding experiments are described in the Supplementary Information (Fig. S22). Two pairs of n-type and p-type moulded disks with the diameter of 4.0 mm and thickness of 1.0 mm were assembled by soldering with a Bi-Sn solder to Cu foil electrodes on an alumina hemisphere (Fig. 5a). The internal resistance was as low as 0.014 Ω , comparable to that of the conventional module. Under the temperature difference of 14 °C, this generator produced an output voltage of 8.0 mV, output power of 1.1 mW, and output power density of 2.3 mW cm^{-2} (Fig. 5b). Furthermore, the predicted power output density on the fitted function with the data points is as high as 26.3 mW cm^{-2} under the temperature difference of 50 °C, which competes on a par with the conventional module⁵³. These results clearly demonstrates the practicability of the painting technology in terms of the TE performance as well as the processability.”

Comment 1: In treatment of uncertainties are missed. It is no information provided about the repeatability. There is no statistics of the result data found, even repeatability is not described.

Response: To address the uncertainty issue, we characterized the thermoelectric properties of more than three sets of n-type and p-type painted samples and added the standard errors to each data

points in Fig. 2 in the revised manuscript. The uncertainties of the electrical conductivity, Seebeck coefficients, and thermal conductivity were 1.5 %, 1.0%, and 5.9%, respectively, which demonstrates the reproducibility of the painting technology. The average ZT values of the n- and p-type samples at room temperature marked 0.51 and 1.09 respectively (Fig. 2g), where the maximum values reached 0.67 for the n-type sample and 1.21 for the p-type sample at 100 °C (Fig. 2g).

We also repeatedly measured the output power characteristics of the painted generators by more than three times and added error bars in each data points in Fig 3 and 4 in the revised manuscript.

The data shows the highly reproducible power output on even curved surfaces.

Comment 2: Authors are strongly recommended to provide a design that could have a comparable performance of conventional off-the-shelf flat thermoelectric generators and with the ability of match any curved surfaces. Presented performance does not make sense otherwise.

Response: Please see the above response for the comment 1.

▪ **Reviewer #3**

General comment: The paper presents shape-engineerable thermoelectric modules which can minimize heat loss caused by the incompatible contact between the heat sources and the thermoelectric modules. The thought is very important for practical applications. However, what distinguish the paper are the scientific spots and the proof of arguments. Here are a few observations.

Comment 1: The authors argue that the painted n-type and for p-type legs have high ZT values of 0.69 and 1.15, respectively. These results seem to be sound. However, the major gain of the high ZT values is from the ultralow thermal conductivity (only 0.5-0.6 W m⁻¹ K⁻¹). One can separate the contribution of lattice thermal conductivity which is only 0.1-0.2 W m⁻¹ K⁻¹, apart from the contribution of electronic component. This ultralow lattice thermal conductivity is not convincing because the theoretical minimum thermal conductivity of binary Bi₂Te₃ is about 0.3 W m⁻¹ K⁻¹.

Comment 2: The explanation of the ultralow lattice thermal conductivities provided is not reasonable. From the microstructure of sintered materials as shown in Figure 2, the well-sintered grains are in micrometre size. Therefore, the major phonon scattering of the materials should be phonon-phonon interaction rather than boundary scattering in the materials as given in the manuscript. In addition, there is no experimental evidence of the existence of negative curvature boundaries. An additional HRTEM experiment is necessary to observe the negative curvature boundaries.

Response: We agree that the bulk-level ZT values of the painted materials originate from the ultralow thermal conductivity, which cannot be explained solely by grain boundary scattering mechanism.

As the reviewer commented, to convince the thermal conductivities and ZT values, we characterized the thermoelectric properties of more than three sets of n-type and p-type samples, as shown in Fig. 2 in the revised manuscript. The thermal conductivity are reliably reproduced on every sample with the uncertainties of 5.9%, which demonstrates the ultralow thermal conductivities are not in fact the measurement artefact but the actual properties of materials.

The thermal conductivity is the sum of electronic and phonon contribution, which is called the lattice thermal conductivity (κ_l). The lattice thermal conductivity can be calculated by subtracting the electronic contribution to the thermal conductivity (κ_e) from total thermal conductivity (κ), which was estimated by using the Wiedemann-Franz Law ($\kappa_e = LT\sigma$, where L is the Lorenz number, T is the absolute temperature, σ is the electrical conductivity). The Lorenz number of $2.0 \times 10^{-8} \text{ V}^2 \text{ K}^{-2}$ is the typically used value for a degenerate semiconductor. However, in the recently published papers, the more reliable calculated value of $\sim 1.6 \times 10^{-8} \text{ V}^2 \text{ K}^{-2}$ was widely used in Bi₂Te₃ related materials^{A3-A6}. Based on this value, the minimum calculated κ_l was 0.19 W m⁻¹ K⁻¹ for n-type and 0.20 W m⁻¹ K⁻¹ for p-type painted

materials. As the reviewer commented, these values are lower or comparable than the predicted minimum κ_1 of $0.31 \text{ W m}^{-1} \text{ K}^{-1}$ in n-type Bi_2Te_3 and $0.20 \text{ W m}^{-1} \text{ K}^{-1}$ and p-type $(\text{Bi,Sb})_2\text{Te}_3$ ^{A7}. However, **we would like to point out that these predicted minima are calculated with the assumption of materials with full densities, using the Debye-Callaway model, whereas the densities of painted materials are at most 50~55% of full densities.**

Generally, phonon scattering can be explained by several mechanisms: Umklapp phonon-phonon scattering, phonon-impurity scattering, phonon-electron scattering, phonon-boundary scattering. The improvement of ZT values in nanostructured thermoelectric materials originates in the reduction of thermal conductivity by phonon-grain boundary scattering arising from the increased interface density. As the reviewer commented, the grain sizes of the painted samples range from several hundred nanometres to several micrometres so that their ultralow lattice thermal conductivities are difficult to be explained solely by grain boundary scattering.

One possible explanation for the ultralow lattice thermal conductivity is the porosity of materials because the porosity in the painted samples reach 45~50%, which suggest that the phonon scattering at multiple pore sites can be the major contribution to the reduced thermal conductivity.

To analyse the porosity of painted samples, we conducted the Brunauer Emmett and Teller (BET) measurement (Fig. A7). Both n-type and p-type samples have pores with the size less than 50 nm, which should be located at the interfaces formed from Sb_2Te_4 sintering aids rather than grains insides. These small pores can act as scattering sites for phonons with short wavelengths. However, the volumes of these small pores are responsible for only 2~3% porosity in the painted samples, calculated based on the measured pore volumes, which suggests the existence of micro-scale pores.

Figure A7 | Nitrogen adsorption-desorption isotherms of the painted samples. (a) n-type and (b) p-type samples. The inset shows the pore size distributions.

To confirm the micro-scale pores, the microstructure of the painted samples was analysed by the SEM. As shown in Fig. A8, the multiple pores with the size ranging from several tens of nanometres to several microns were clearly observed in the SEM images. **Given that the presence of multi-scale pores can reduce the thermal conductivity by phonon scattering with a broad range of wavelength at pore sites, the ultralow thermal conductivities of painted TE materials can be explained by the porosity.**

Figure A8 | Low-magnification SEM images of the painted (a) n-type and (b) p-type samples. The red circles show the micro-scale pores in the samples.

To further quantitatively estimate the porosity effect on the thermal transport, the κ_l of painted samples were corrected using the modified formulation (eqn. (4)) of the effective medium theory suggested by Lee *et al.*^{A8}:

$$\kappa_l = \kappa_h \frac{(2-2\Phi)}{(2+\Phi)} \quad (4)$$

, where κ_h and Φ is the lattice thermal conductivity of host materials and the porosity respectively. The calculated minimum κ_l of the n-type and p-type painted samples are **0.44 W m⁻¹ K⁻¹** and **0.47 W m⁻¹ K⁻¹** (Fig. A9) respectively, **which are comparable to those of typical nanostructured bulk materials prepared from ball-milled Bi₂(Te,Se)₃ and (Bi,Sb)₂Te₃.** Consequently, these results suggest that the porosity of painted samples can be a major contribution to the reduced latticed thermal conductivity for boundary scattering of phonons at pore sites, rather than grain boundaries.

Figure A9 | Calculated lattice thermal conductivities of n-type and p-type painted samples using the modified formulation (eqn. (4)) of the effective medium theory^{A8}.

We added the BET isotherms, the SEM images, and the calculated lattice thermal conductivity in the revised Supplementary Information (Fig. S11-S13), removed the sentences on negative curvature boundaries, and included the following sentences in the revised manuscript (page 9). Also, the experimental details for the above measurements were included in the revised Supplementary Information.

“The calculated lattice thermal conductivities were as low as $0.19 \text{ W m}^{-1} \text{ K}^{-1}$ for n-type and $0.20 \text{ W m}^{-1} \text{ K}^{-1}$ for p-type painted materials (Fig. S11). These values are lower or comparable than the predicted minimum κ_l of $0.31 \text{ W m}^{-1} \text{ K}^{-1}$ in n-type Bi_2Te_3 and $0.20 \text{ W m}^{-1} \text{ K}^{-1}$ and p-type $(\text{Bi,Sb})_2\text{Te}_3$, which is calculated using the Debye-Callaway model with the assumption of full densities⁴⁹. One possible explanation for the ultralow lattice thermal conductivity is the porosity of materials. Although the Sb_2Te_3 ChaM promotes the sintering of TE particles, their densities are still lower than the bulk values of $6.5\sim 7.5 \text{ g/cm}^3$. The analysis of porosity with N_2 adsorption measurement and SEM of these materials (Fig. S12 and S13) reveal the existence of both nano-scale and micro-scale pores. These multi-scale pores can significantly reduce the thermal conductivity by phonon scattering with a broad range of wavelength at pore sites. To further quantitatively estimate the porosity effect on the thermal transport, the κ_l of painted samples were corrected by using the modified formulation of the effective medium theory suggested by Lee *et al.*⁵⁰: $\kappa_l = \kappa_h \frac{(2-2\Phi)}{(2+\Phi)}$, where κ_h and Φ are the lattice thermal conductivity of host materials and the porosity respectively. The calculated minimum κ_l of the n-type and p-type painted samples are $0.44 \text{ W m}^{-1} \text{ K}^{-1}$ and $0.47 \text{ W m}^{-1} \text{ K}^{-1}$ (Fig. S11) respectively, which are comparable to those of typical nanostructured bulk materials prepared from ball-milled $\text{Bi}_2(\text{Te,Se})_3$ and $(\text{Bi,Sb})_2\text{Te}_3$.”

Comment 3: One of the main achievements of this work is using molecular Sb_2Te_3 chalcogenidometalate as a sintering aid to improve the density of the painted thick films hence performance. The most interesting work one expects to see is the sintering mechanism during the aided sintering process. This work is extremely important from the scientific point of view, but is missing in the manuscript.

Response: We agree that a proper understanding of sintering mechanism is crucial in extending versatility of the currently developed technique. As manifested from the microstructures (Fig. 2a-d in the revised manuscript), the grain morphology clearly dictates that the grain growth took place in a layer-by-layer mode, which requires 2-dimensional nucleation event from a liquid medium as a prerequisite^{A9}. **This implies that the added sintering aid formed a liquid phase at the sintering temperature, which provides a diffusion path for grain growth.** As evidenced by the DSC curves of n-type and p-type paints in Fig. S7 in the revised Supplementary Information, the Te phase formed from the Sb_2Te_3 Cham sintering aid is melted at ~ 420 °C, lower than the sintering temperature of 450 °C. It means that the liquefied Te can contribute to the sintering upon heat treatment. A possible contribution from the viscous flow mechanism during the initial stage of the liquid phase sintering was ruled out based on an analysis on a time-dependent shrinkage measurement as shown in Fig. A10, where the time exponent of 0.08 is determined to be much smaller than the theoretically expected one.

Figure A10 | A shrinkage vs. time plot of the n-type paint during sintering at 450 °C.

It is noted that the viscous flow mechanism during liquid-phase sintering is often represented as the

following relation^{A10}:

$$\frac{\Delta l}{l} \propto t^{1+y} \quad (5)$$

where l and t denote a linear dimension of the specimen and sintering time, respectively. Here, the exponent $1 + y$ is slightly larger than unity due to increasing driving force with decreasing pore size during the process.

We replaced the time-dependent density graph with the plot of a shrinkage versus time (Fig. S8), and included the following paragraph in the revised manuscript (page 7)

“As manifested from the microstructures (Fig. 2a-d), the grain morphology clearly dictates that the grain growth took place in a layer-by-layer mode, which requires 2-dimensional nucleation event from a liquid medium as a prerequisite⁴⁴. This implies that the added sintering aid formed a liquid phase at the sintering temperature, which provides a diffusion path for grain growth. As evidenced by the DSC curves of n-type and p-type paints (Fig. S7), the Te phase formed from the Sb₂Te₃ ChaM sintering aid is melted at ~420 °C, lower than the sintering temperature of 450 °C. It means that the liquefied Te can contribute to the liquid phase sintering upon heat treatment. A possible contribution from the viscous flow mechanism during the initial stage of the liquid phase sintering was ruled out based on an analysis on a time-dependent shrinkage measurement as shown in Fig. S8, where the time exponent of 0.08 is determined to be much smaller than the theoretically expected one. It is noted that the viscous flow mechanism during liquid-phase sintering is often represented as the following relation⁴⁵: $\Delta l/l \propto t^{1+y}$, where l and t denote a linear dimension of the sample and sintering time, respectively. Here, the exponent $1 + y$ is slightly larger than unity due to increasing driving force with decreasing pore size during the process.”

Comment 4: The authors reported that the painted films on the hemispherical alumina substrate show unexpectedly low performance, which has been attributed to the measurement artifact. However, this explanation is a sort of debatable. The heat loss from the highly thermally conducting alumina substrate should be taken into consideration.

Response: We agree that the heat loss from an alumina substrate can jeopardize the power generating performance of the painted hemispherical generator. However, considering the TE legs near the apex of the alumina hemisphere are very close to a planar heat source during the measurement, it is difficult to rule out the possibility that the radiation and convection influence the measurement of power output. To minimise the radiation and convection during the measurement, we designed the measurement set-up where the planar heat source was fully covered with a glass fabric and the apex of the hemispherical generator was thermally connected by thermal pads, as shown in Fig. A11a. The power output of the

hemispherical generator was comparatively measured with and without a glass fabric covering. The output voltage and output power density obtained with a glass fabric covering are ~20% and ~40% higher than those measured without it (Fig. A11b). Assuming the same dimension of TE legs with those of other painted devices, the plotted output power density of this device was significantly increased from the previous data, (Fig. A12), which demonstrates that the radiation and convection was one of the major factors to decrease the performance. However, the data from the hemispherical generator still deviate from others, indicating that this discrepancy should originate from the heat loss from a thermally conducting alumina substrate, as the reviewer suggested.

Figure A11 | Measurement of the power output of hemispherical TE generator. (a) Scheme of the TE power measurement set-up without glass fabric and with glass fabric to minimize convection and radiation from the hot plate to TE legs. (b) Comparison of output voltage and power densities of hemispherical TE generator measured with and without a glass fabric.

Figure A12 | Comparison of output power densities of the painted TE devices. The output power density of the hemispherical generator was calculated with the assumption of same dimensions of TE legs with others.

We replaced the previous output characteristics of the painted hemispherical generator and the measurement scheme with new data measured with a glass fiber covering (Fig. 4 and Fig. S16) and new scheme, respectively (Fig. S17). The paragraph related to the painted hemispherical generator were replaced with the following paragraph in the revised manuscript (page 11-12)

“To further demonstrate the processability of TE painting onto large-sized curved surfaces with a full coverage, TE device was fabricated on a hemispherical alumina substrate with the diameter of ~ 70 mm (Fig. 4g). We introduced 5.5 couples of triangular TE layers with ~ 15 mm at the base and 25 mm in height and obtained the internal resistance of 40.2Ω , which is expected for the enlarged TE layers (67 % higher aspect ratio). To minimise radiation or convection factor from a heat source, the planar heat source was fully covered with a glass fabric and the apex of the hemispherical generator was thermally connected by thermal pads (Fig. S17). Exposed to a temperature difference of $20.1 \text{ }^\circ\text{C}$, this device produced the output voltage of 22.5 mV, the output power of $3.0 \mu\text{W}$, and the output power density of 0.073 mW cm^{-2} (Fig. 4h), which are significantly lower than those of the other devices. It is understood that this low output power density can originate in the longer TE legs which increase the internal resistance since the output power density is inversely proportional to the leg length under same temperature difference⁵². Assuming the identical dimension of the TE legs to those of other devices, the plotted output power density approached towards the others (Fig. S16). As evidenced by the lower output voltage, small deviation in the graph (Fig. S16) can be due to the heat loss from a thermally conducting alumina substrate, which forces the external temperatures to be different from the actual temperature applied to the TE legs.”

Comment 5: The large internal resistance of this device as high as 25.8 ohm suggests that the contact resistance between the electrodes and TE legs might be considerably large, according to the high electrical conductivity of the legs. However, no further analysis was provided at this point.

Response: We agree that the significant contact resistance cause the high internal resistance of the painted generator. We measured the contact resistance between the electrode deposited with Ag paint and the painted TE leg by the transmission line method (Fig. A13). The measured contact resistance is as high as $4.8 \times 10^{-2} \Omega \text{ cm}^2$, which is more than four orders of magnitude higher than the typical contact resistance of Bi_2Te_3 -based materials with metal electrodes^{A11}. This high contact resistance should be responsible for the relatively high internal resistance of the painted TE generator.

Figure A13 | Contact resistance measurement by the transmission line method. From the linear fit of resistance versus distance between patterns, the specific contact resistance was evaluated.

We added **the plot of the resistance versus the distance** in the revised Supplementary Information (Figure S14), and **replaced the sentences related to the resistance of the painted generator with the following sentences** in the revised manuscript (page 10)

“The internal resistance of this device was 25.8 Ω , higher than the expected resistance in reference to the electrical properties, suggesting that the contact resistance between the Ag electrode and the TE leg is considerably high. We measured the contact resistance between the Ag electrode and the painted TE leg by the transmission line method (Fig. S14). The measured contact resistance is quite high at $4.8 \times 10^{-2} \Omega \text{ cm}^2$, which is three or four orders of magnitude higher than the contact resistance observed in conventional module composed of Bi_2Te_3 -based TE legs⁵¹ and can be responsible for the internal resistance of the painted TE generator.”

References

- A1. Yazawa, K., Shakouri, A. Optimization of power and efficiency of thermoelectric devices with asymmetric thermal contacts *J. Appl. Phys.* **111**, 024509 (2012).
- A2. Rowe, D. M., Min, G. Evaluation of thermoelectric modules for power generation. *J. Power Sources* **73**, 193-198 (1998).
- A3. Kim, S. I. *et al.* Dense dislocation arrays embedded in grain boundaries for high-performance bulk thermoelectrics. *Science* **348**, 109-114 (2015).
- A4. Zhang, Y. *et al.* Hot carrier filtering in solution processed heterostructures: a paradigm for improving thermoelectric efficiency. *Adv. Mater.* **26**, 2755-2761 (2014).
- A5. Hu, L. *et al.* Tuning multiscale microstructures to enhance thermoelectric performance of n-type bismuth-telluride-based solid solutions. *Adv. Energy Mater.* **5**, 1500411 (2015)
- A6. Yan, X. *et al.* Experimental studies on anisotropic thermoelectric properties and structures of n-type $\text{Bi}_2\text{Te}_{2.7}\text{Se}_{0.3}$. *Nano Lett.* **10**, 3373-3378 (2010)
- A7. Chiritescu, C., Mortensen, C., Cahill, D. G., Johnson, D., Zschack, P. Lower limit to the lattice thermal conductivity of nanostructured Bi_2Te_3 -based materials *J. Appl. Phys.* **106**, 073503 (2009).
- A8. Lee, H., *et al.* Effects of nanoscale porosity on thermoelectric properties of SiGe. *J. Appl. Phys.* **107**, 094308 (2010).
- A9. Jo, W., Kim, D. -Y., Hwang, N. -M. Effect of interface structure on the microstructural evolution of ceramics. *J. Am. Ceram. Soc.* **89**, 2369-2380 (2006).
- A10. Kingery, W. D. Densification during sintering in the presence of a liquid phase. I. Theory. *J. Appl. Phys.* **30**, 301-306 (1959).
- A11. Gupta, R. P., Mccarty, R., F, Sharp, J. Practical contact resistance measurement method for bulk Bi_2Te_3 -based thermoelectric devices. *J. Electron. Mater.* **43**, 1608-1612 (2014)

Reviewer #2 (Remarks to the Author)

The addition of numerical analysis makes a lot of sense and explains the arguments. However, I just want authors to make clear if the heat losses are included in the calculations. If not, I would suggest to show the impact in the article. I think only by FEA simulation would be fine.

Reviewer #3 (Remarks to the Author)

I have read the revised manuscript and point-by-point response letter. The authors treated my questions but gave dissatisfactory answers. The reliable and reproducible data are extremely important for high international impact journals such as Nature Communications. However, I intensively feel that partial experimental data in the revised manuscript are unbelievable.

(1) Authors consider that one possible explanation for the ultralow lattice thermal conductivity is the porosity of materials because the porosity in the painted samples reach 45~50% (to see P12 of Rebuttal file). Here, the porosity is a very important data. However, the important data does not appear in article file and supplementary file. Authors should give an explanation.

(2) To further quantitatively estimate the porosity effect on the thermal transport, the κ_l of painted samples were corrected by using the modified formulation of the effective medium theory suggested by Lee et al.: $\kappa_l = \kappa_h(2 - 2\Phi)(2 + \Phi)$, where κ_h and Φ are the lattice thermal conductivity of host materials and the porosity respectively. According to the calculated data listed in the revised manuscript ($\kappa_l = 0.19$ for n-type and 0.20 for p-type, $\kappa_h = 0.44$ for n-type and 0.47 for p-type), the Φ should be about 60% for n-type painted sample and 62% for p-type painted sample, which are much more than 45~50%. Why?

(3) There are a large number of reports about the effect of nanopores and micropores on the electrical and thermal transport properties of materials (J. Appl. Phys. 2007, 101, 014322; Appl. Phys. Lett. 2007, 91, 223110; Appl. Phys. Lett. 2004, 84, 1885; Appl. Phys. Lett. 2004, 84, 687; Acta Mater. 2012, 60, 1741; Appl. Phys. Lett. 2014, 104, 142104; J. Appl. Phys. 2012, 112, 044305; Appl. Phys. Lett. 2008, 93, 064302). Since the porous structure easily results in the remarkable decrease in the electrical conductivity, the porosity is generally less than 20%. In the revised manuscript, the porosity of the painted samples is extremely high and more than 50%; however, the electrical conductivity is very close to that of the bulk materials. Authors have not given any explanation of the fancy phenomenon.

(4) As shown in Fig. 2a-d, it is very hard to understand that the grain growth took place in a layer-by-layer mode. I intensively suggest that the author observes the microstructures of cross-section fracture for the painted samples.

(5) To explain why the painted samples with extremely high porosity have excellent electrical transport properties, an additional HRTEM experiment is very necessary to observe the microstructures on the nanometer scale.

Response to the reviewers' comments

The followings are the responses to the reviewers' comments for the manuscript "High performance shape-engineerable thermoelectric painting."

▪ Reviewer #2

General comment: The addition of numerical analysis makes a lot of sense and explains the arguments. However, I just want authors to make clear if the heat losses are included in the calculations. If not, I would suggest to show the impact in the article. I think only by FEA simulation would be fine.

Response: We considered **the heat loss in the FEM by including the convective heat transfer**. To simulate the natural convection over all the surfaces that are exposed to air, the convection heat transfer coefficient was $10 \text{ W m}^{-2} \text{ K}^{-1}$ with an ambient temperature of $25 \text{ }^\circ\text{C}$. These values are generally used to simulate the natural convection^{A1}.

To clarify we included **the following sentence** in the revised manuscript (page 14).

“The heat loss in the FEM was considered by including the convective heat transfer. To simulate the natural convection over all the surfaces that are exposed to air, the convection heat transfer coefficient was $10 \text{ W m}^{-2} \text{ K}^{-1}$ with an ambient temperature of $25 \text{ }^\circ\text{C}$.”

▪ Reviewer #3

General comment: I have read the revised manuscript and point-by-point response letter. The authors treated my questions but gave dissatisfactory answers. The reliable and reproducible data are extremely important for high international impact journals such as Nature Communications. However, I intensively feel that partial experimental data in the revised manuscript are unbelievable.

Response: We agree that the reliable and reproducible data are essential to validate the feasibility of the currently developed technique. We have tried our best to obtain the reliable and reproducible thermoelectric properties and the data shown in Fig. 2g-j are the average values obtained by the measurement on more than three sets of the painted samples. The uncertainties of the electrical conductivity, Seebeck coefficients, and thermal conductivity were 1.5 %, 1.0%, and 5.9%, respectively, which demonstrates the reproducibility of the painting technology.

Comment 1: Authors consider that one possible explanation for the ultralow lattice thermal conductivity is the porosity of materials because the porosity in the painted samples reach 45~50% (to see P12 of Rebuttal file). Here, the porosity is a very important data. However, the important data does not appear in article file and supplementary file. Authors should give an explanation.

Response: We agree that the porosity data is crucial to explain the ultralow thermal conductivity of the painted samples. Generally, the porosity is defined as a measure of the void spaces in a material, and is a fraction of the volume of voids over the total volume. The overall porosity of a solid material can be estimated by a direct method of comparing the sample volume to the theoretical volume of the materials with no pores. In the current study, we **evaluated the porosity of the samples by this method using the data of the sample densities in Fig. 2e in the revised manuscript, compared with the bulk densities of $\text{Bi}_{2.0}\text{Te}_{2.7}\text{Se}_{0.3}$ (7.55 g cm^{-3}) for the n-type and $\text{Bi}_{0.4}\text{Sb}_{1.6}\text{Te}_{3.0}$ (6.785 g cm^{-3}) for the p-type materials.** For example, the n-type painted sample exhibited the density of 3.85 g cm^{-3} , from which the porosity of 0.47 was evaluated by the calculation of $1 - \frac{3.85 \text{ g cm}^{-3}}{7.55 \text{ g cm}^{-3}}$.

We included **the porosity data** and **the following sentences** in the revised manuscript (page 9)

“The overall porosities of the painted samples were estimated by the direct method of comparing the sample density to the theoretical density of bulk materials with identical compositions. Using the bulk densities of $\text{Bi}_{2.0}\text{Te}_{2.7}\text{Se}_{0.3}$ (7.55 g cm^{-3}) for the n-type and $\text{Bi}_{0.4}\text{Sb}_{1.6}\text{Te}_{3.0}$ (6.785 g cm^{-3}) for the p-type materials, the calculated porosities of the painted samples were 0.47 for the n-type and 0.46 for the p-

type samples.”

Comment 2: To further quantitatively estimate the porosity effect on the thermal transport, the κ_l of painted samples were corrected by using the modified formulation of the effective medium theory suggested by Lee et al.: $\kappa_l = \kappa_h(2 - 2\Phi)(2 + \Phi)$, where κ_h and Φ are the lattice thermal conductivity of host materials and the porosity respectively. According to the calculated data listed in the revised manuscript ($\kappa_l = 0.19$ for n-type and 0.20 for p-type, $\kappa_h = 0.44$ for n-type and 0.47 for p-type), the Φ should be about 60% for n-type painted sample and 62% for p-type painted sample, which are much more than 45~50%. Why?

Response: We appreciate the great effort made by the reviewer. It seems that there was a slight confusion by the reviewer in evaluating the lattice thermal conductivity using the modified formulation of the effective medium theory for our case. In order to confirm the accuracy of our approach, we have checked the calculation in several different ways and we reached to the same exact conclusion. Please refer to the details of the calculation below.

The calculated porosities and the measured lattice thermal conductivities of the painted samples were summarized in Table A1.

Table A1 | Porosity and lattice thermal conductivity of the painted materials

	Porosity	Lattice thermal conductivity ($\text{W m}^{-1} \text{K}^{-1}$)
N-type	0.462	0.193
P-type	0.470	0.202

Based on the modified formulation of the effective medium theory^{A2}, **the following equation was derived with the porosity data and the lattice thermal conductivity of the n-type sample:**

$$0.193 \text{ W m}^{-1} \text{ K}^{-1} = \kappa_h \frac{(2 - 2 \times 0.462)}{(2 + 0.462)}$$

The estimated lattice thermal conductivity of the sample with full density was $\kappa_h = 0.442 \text{ W m}^{-1} \text{ K}^{-1}$.¹ For the p-type sample, the corrected lattice thermal conductivity was $\kappa_h = 0.471 \text{ W m}^{-1} \text{ K}^{-1}$.

Comment 3: There are a large number of reports about the effect of nanopores and micropores on the electrical and thermal transport properties of materials (J. Appl. Phys. 2007, 101, 014322; Appl. Phys. Lett. 2007, 91, 223110; Appl. Phys. Lett. 2004, 84, 1885; Appl. Phys. Lett. 2004, 84, 687; Acta Mater. 2012, 60, 1741; Appl. Phys. Lett. 2014, 104, 142104; J. Appl. Phys. 2012, 112, 044305; Appl. Phys.

Lett. 2008, 93, 064302). Since the porous structure easily results in the remarkable decrease in the electrical conductivity, the porosity is generally less than 20%. In the revised manuscript, the porosity of the painted samples is extremely high and more than 50%; however, the electrical conductivity is very close to that of the bulk materials. Authors have not given any explanation of the fancy phenomenon.

Response: We agree that a proper understanding of the high electrical conductivity is essential to extend the versatility of the currently developed technology. To understand the charge transport of the painted materials, one should classify the factors to determine the electrical conductivity as the electrical conductivity (σ) is expressed by $\sigma = en\mu$, where e is the electron charge, n is the carrier concentration and μ is the carrier mobility.

As the reviewer commented, the porosity of solid materials decreases an electrical conductivity due to scattering of charge carriers at the pore sites^{A2}. When a charge carrier passes near a pore, it is scattered to a different wave vector due to the potential perturbation. This charge carrier scattering generally degrade the carrier mobility, rather than the carrier concentration. The carrier scattering effect on mobility can be qualitatively described by the empirical Matthiessen's rule^{A3}

$$\frac{1}{\mu_{tot}} = \frac{1}{\mu_{bulk}} + \frac{1}{\mu_{impurity}} + \frac{1}{\mu_{boundary}} + \frac{1}{\mu_{pore}} \quad (1)$$

The total scattering mechanism can be expressed as the sum of the contribution from different electron scattering process. For example, μ_{bulk} is the mobility determined solely by the scattering on acoustic phonons. In the painted materials, considering no additional impurity element except Bi, Sb, and Te, $\mu_{boundary}$ and μ_{pore} should be the critical factors to determine μ_{tot} . Lee et al. intensively studied the charge transport of thermoelectric materials depending on the porosity and the grain size by numerical simulation and they suggested that **the porosity effect on electrical properties become weaker for larger grains** (Fig. A1)^{A2}. Since **the material with larger grains necessarily has larger pores with the lower number density under same porosity, the scattering rate is lower and the mobility is higher for larger grain sizes**. In fact, we observed the lower mobility of 149 cm² V⁻¹ s⁻¹ for the n-type and 141 cm² V⁻¹ s⁻¹ for the p-type painted materials, compared with bulk values (400~1200 cm² V⁻¹ s⁻¹ in single crystal and 250~350 cm² V⁻¹ s⁻¹ in polycrystal)^{A4,A5}, which can be attributed to the porosity effect. However, these values are still identical order of magnitude to those of bulk materials. **The fact that the grain size is in the range of several micrometres and the pores are mainly macro-scale (less than 3% of micro-pores in volume) in the painted materials (Fig. A2) suggests that the relatively high mobility is attributed to the lower number density of the pore.**

Figure A1 | Electrical conductivity of SiGe as a function of porosity for different grain sizes^{A2}.

Figure A2 | Low-magnification cross-sectional SEM image of the painted materials.

Another important factor to determine the electrical conductivity is the carrier concentration. To overcome the lower mobility of the painted samples than those of bulk, we chose the composition of $\text{Bi}_{0.4}\text{Sb}_{1.6}\text{Te}_{3.0}$ (p-type) and $\text{Bi}_{2.0}\text{Te}_{2.7}\text{Se}_{0.3}$ (n-type) for host matrix materials. These are not typically utilized composition for the highest thermoelectric performance. However, they are known to exhibit higher carrier concentration by the formation of Sb_{Te} antisite defect to provide a hole in p-type and Se vacancy defect to provide an electron in n-type. In fact, we observed two or three-fold higher carrier concentration of $3.0 \times 10^{19} \text{ cm}^{-3}$ for the n-type and $2.9 \times 10^{19} \text{ cm}^{-3}$ for the p-type painted materials, compared with $1\text{--}2 \text{ cm}^{-3}$ of typically used materials ($\text{Bi}_{0.5}\text{Sb}_{1.5}\text{Te}_{3.0}$ for p-type and Bi_2Te_3 for n-type). Although these high n decreased the Seebeck coefficient since they are reciprocally proportional, the electrical conductivities were significantly increased up to 650~750 S/cm at room temperature, close to bulk values. Consequently, in spite of high porosity, the high carrier concentration, the low number density of pores and bulk-scale grains can result in the high electrical conductivity of the painted materials.

We included **the following sentences** in the revised manuscript (page 8 and 10-11) and **replaced**

Fig.S13 with a new SEM image (Fig. A2).

“since the electrical conductivity (σ) is expressed by $\sigma=en\mu$, where e is the electron charge, n is the carrier concentration and μ is the carrier mobility.” (page 8)

“Generally, the porosity of solid materials strongly affects the charge carrier transport due to scattering of carriers at the pore sites. A charge carrier passing near a pore is scattered due to the potential perturbation⁵⁰, degrading the carrier mobility and eventually the electrical conductivity. The carrier scattering effect on mobility can be qualitatively described by the Matthiessen’s rule⁵¹

$$\frac{1}{\mu_{\text{tot}}} = \frac{1}{\mu_{\text{bulk}}} + \frac{1}{\mu_{\text{impurity}}} + \frac{1}{\mu_{\text{boundary}}} + \frac{1}{\mu_{\text{pore}}} \quad (1)$$

Accordingly, the total scattering is the sum of the contribution of different carrier scattering mechanism. For example, μ_{bulk} is the mobility induced solely by the carrier scattering with acoustic phonons. In the painted materials, considering no additional impurity element except Bi, Sb and Te, μ_{boundary} and μ_{pore} should be the critical factors to determine the overall mobility. Lee *et al.* suggested that the porosity effect on electrical properties become weaker for larger grains⁵⁰. Since the material with larger grains necessarily has larger pores with the lower number density under the same porosity, the scattering rate is reduced and mobility is enhanced for larger grain sizes. The fact that the grain size is in the range of several micrometres (Fig. 2a and 2c) and the pores are mainly macro-scale in the painted materials (less than 3% of micro-pores in volume) suggests that the moderately high mobility is attributed to the lower number density of the pore.

Another important factor to determine the electrical conductivity is the carrier concentration. To overcome the lower mobility of the painted samples than those of bulk, we chose the composition of $\text{Bi}_{0.4}\text{Sb}_{1.6}\text{Te}_{3.0}$ (p-type) and $\text{Bi}_{2.0}\text{Te}_{2.7}\text{Se}_{0.3}$ (n-type) for host matrix materials. The materials with such compositions are known to exhibit high carrier concentration by the formation of Sb_{Te} antisite defect to provide hole in p-type and Se vacancy defect to provide electron in n-type. In fact, the carrier concentrations of the painted samples were two or three-fold higher than $1\sim 2 \text{ cm}^{-3}$ of typically used Bi_2Te_3 based materials². Although these high carrier concentrations decreased the Seebeck coefficients, the electrical conductivities were significantly increased up to $650\text{--}750 \text{ S cm}^{-1}$ at room temperature, close to bulk values. Consequently, in spite of high porosity, the high carrier concentration, the low number density of pores, and bulk-scale grains can result in the high electrical conductivity of the painted materials.” (page 10-11)

Comment 4: As shown in Fig. 2a-d, it is very hard to understand that the grain growth took place in a layer-by-layer mode. I intensively suggest that the author observes the microstructures of cross-

section fracture for the painted samples.

Response: As presented in in the SEM image (Fig. A4), which was taken from a fractured surface, **the growth mode of the current sample is obviously a nucleation and lateral growth**, i.e., layer-by-layer growth. The red circles in the SEM image show the stereotypical microstructure formed by a nucleation and lateral growth.

Figure A4 | SEM image of the fractured surface of the n-type painted material. The red circles show the stereotypical microstructure formed by a nucleation and lateral growth.

To avoid any possible confusion, we added the SEM image of a fractured surface in the revised Supplementary Information (Supplementary Fig. 8) and included the following sentences in the revised manuscript (page 7).

“The scanning electron microscope (SEM) image of the fractured surface (Supplementary Fig. 8) shows the stereotypical microstructure formed by a nucleation and lateral growth⁴⁴.”

Comment 5: To explain why the painted samples with extremely high porosity have excellent electrical transport properties, an additional HRTEM experiment is very necessary to observe the microstructures on the nanometer scale.

Response: The high electrical properties of the painted materials originate from the electrical conductivity, rather than the Seebeck coefficient. Generally, **nanoscale structures or defects in thermoelectric materials which can increase the electrical conductivity have been very rarely suggested because they usually act as carrier scattering sites to degrade the electrical conductivity**. To the best of our knowledge, only few reports have suggested such nanostructures to increase the electrical conductivity. For example, Yu, et al. reported the modulation doping concept

where the heavily doped and isolated nanograins can increase the overall carrier mobility and electrical conductivity^{A6}.

We conducted HRTEM analysis of the painted materials and identified two regions formed from host materials (grain inside) and formed from the Sb_2Te_3 sintering aide (interfaces or boundaries). As shown in Fig.A5a, inside grains we could not find any interesting microstructures which can influence the electrical properties. **Near boundaries, multiple nanoscale grains and precipitates were observed (Fig.A5b)**. These nanoscale structures could be formed from the Sb_2Te_3 sintering aide while they were integrated into grains during sintering. However, these nanostructures seem not to be heavily doped because the EDS elemental analysis did not show the different composition of this region from host grains. Generally, the doping of Bi_2Te_3 based materials is controlled by the formation of defects such as antisite and vacancy, which typically accompanies the change of local composition. Thus, we can rule out the possibility of modulation doping effect in the painted materials. **Although these nanostructures can enhance the Seebeck coefficient by the energy filtering effect as well as scattering phonons to reduce the thermal conductivity, it is difficult to say that they are one of the reasons for the high electrical conductivity**.

The high electrical conductivity of the painted sample could originate in macrostructures rather than nanostructures. As mentioned in the response for the comment 3, the macro-scale grains and pores resulted in the lower number density of pores, which should scatter less charge carriers than typical nanostructured materials. Therefore, it enabled to maintain the moderately high carrier mobility and electrical conductivity of the painted materials. **Although this interesting phenomenon in the electrical properties of the painted materials are not fully understood in the current study, we believe that the microstructural characterization of the current materials, as suggested by the reviewer, will be an important benchmark result to understand similar paste- or ink-based thermoelectric materials.**

Figure A5 | HRTEM images of the painted materials near (a) grains and (b) boundary.

References

- A1. Incropera, F. P. *Fundamentals of Heat and Mass Transfer*, 7ed.(Willey, 2011)
- A2. Lee, H., *et al.* Effects of nanoscale porosity on thermoelectric properties of SiGe. *J. Appl. Phys.* **107**, 094308 (2010).
- A3. Ashcroft, N. W.; Mermin, N. D. *Solid State Physics* (Holt, Rinehart and Winston, New York, 1976)
- A4. Ivanova, L. D., Granatkina, Y.V. Thermoelectric properties of Bi₂Te₃-Sb₂Te₃ single crystals in the range 100-700 K. *Inorg. Mater.* **36**, 672-677 (2000)
- A5. Rowe, D. M. *Thermoelectric Handbook* (CRC, Boca Raton, 1995).
- A6. Yu, B. *et al.* Enhancement of thermoelectric properties by modulation-doping in silicon germanium alloy nanocomposites. *Nano Lett.* **12**, 2077-2082 (2012)

***Thank you very much for many appropriate and valuable comments.
I am sure that these comments improved the quality of the manuscript significantly.***

Reviewer #3 (Remarks to the Author)

The authors replied my questions by point-by-point pattern. I am satisfied with most of the answers. However, I still have the following questions before I suggest the manuscripts to publish in Nature Communication.

(1) The calculated porosities of the painted samples were 0.47 for the n-type and 0.46 for the p-type samples. The pores are mainly macro-scale in the painted samples (less than 3% of micro-pores in volume). Authors should give a calculated method for "less than 3% of micro-pores in volume".

(2) To overcome the lower mobility of the painted samples than those of bulk, authors chose the composition of $\text{Bi}_{0.4}\text{Sb}_{1.6}\text{Te}_{3.0}$ (p-type) and $\text{Bi}_{2.0}\text{Te}_{2.7}\text{Se}_{0.3}$ (n-type) for host matrix materials and increase the carrier concentration by the formation of SbTe antisite defect to provide hole in p-type and Se vacancy defect to provide electron in n-type. As a result, the carrier concentration of the painted samples reached two or three-fold higher than those of typically used Bi_2Te_3 based materials. Authors should explain why the Seebeck coefficients did not significantly decreased in the case of the carrier concentration increased by 2-3 times. Was this phenomenon reported by other research groups?

(3) Authors firmly convinced that in spite of high porosity up to 40-50%, the painted materials may keep the high electrical conductivity due to the high carrier concentration, the low density of pores. In order to show that these data are reliable, authors should give a theoretical result based on a porous material with the porosity up to 40-50%, not based on porous materials with the porosity less than 20%.

Response to the reviewers' comments

The followings are the responses to the reviewers' comments for the manuscript "High performance shape-engineerable thermoelectric painting."

▪ **Reviewer #3**

General comment: The authors replied my questions by point-by-point pattern. I am satisfied with most of the answers. However, I still have the following questions before I suggest the manuscripts to publish in Nature Communication.

Comment 1: The calculated porosities of the painted samples were 0.47 for the n-type and 0.46 for the p-type samples. The pores are mainly macro-scale in the painted samples (less than 3% of micro-pores in volume). Authors should give a calculated method for "less than 3% of micro-pores in volume".

Response: We appreciate the reviewer's appropriate comment. The porosities of micro-pores in the painted materials were calculated with the micro-pore volumes obtained by N₂ adsorption measurement. The measured micro-pore volumes of the n-type and p-type samples were 0.00261 cm³ g⁻¹ and 0.00319 cm³ g⁻¹ respectively. Also, the average pore sizes were 5.85 nm for the n-type and 5.70 nm for the p-type samples. According to these data and the densities of the samples, the estimated portions of micro-pores in the entire porosity were 2.1% for the n-type sample and 2.5% for the p-type sample.

We included the **following sentences** in the Method section of the revised manuscript.

"The porosities of micro-pores in the painted materials were calculated with the micro-pore volumes obtained by the BET measurement, which were 0.00261 cm³ g⁻¹ for the n-type sample and 0.00319 cm³ g⁻¹ for the p-type sample. Also, the average pore sizes were 5.85 nm for the n-type and 5.70 nm for the p-type samples. According to these data and the densities of the samples, the estimated portions of micro-pores in the entire porosity were 2.1% for the n-type sample and 2.5% for the p-type sample."

Comment 2: To overcome the lower mobility of the painted samples than those of bulk, authors chose the composition of Bi_{0.4}Sb_{1.6}Te_{3.0} (p-type) and Bi_{2.0}Te_{2.7}Se_{0.3} (n-type) for host matrix materials and increase the carrier concentration by the formation of SbTe antisite defect to provide hole in p-type and Se vacancy defect to provide electron in n-type. As a result, the carrier concentration of the painted samples reached two or three-fold higher than those of typically used Bi₂Te₃ based materials.

Authors should explain why the Seebeck coefficients did not significantly decreased in the case of the carrier concentration increased by 2-3 times. Was this phenomenon reported by other research groups?

Response: The materials with the compositions of $\text{Bi}_{0.5}\text{Sb}_{1.5}\text{Te}_{3.0}$ (p-type) and Bi_2Te_3 (n-type), that are widely used for the highest ZT values due to their optimum carrier concentrations ranging $1\sim 2 \times 10^{19} \text{ cm}^{-3}$. These materials typically exhibit the Seebeck coefficients ranging $180\sim 220 \mu\text{V } ^\circ\text{C}^{-1}$ for Bi_2Te_3 (n-type) and $200\sim 240 \mu\text{V } ^\circ\text{C}^{-1}$ for $\text{Bi}_{0.5}\text{Sb}_{1.5}\text{Te}_{3.0}$ (p-type) at room temperature². **Given that the Seebeck coefficient (S) relation of $S \sim n^{-2/3}$, where n is the carrier concentration², the Seebeck coefficient of $114 \mu\text{V } ^\circ\text{C}^{-1}$ for the n-type sample and $171 \mu\text{V } ^\circ\text{C}^{-1}$ for the p-type sample at room temperature are in the appropriate ranges, which is in line with the theoretical relation.**

Comment 3: Authors firmly convinced that in spite of high porosity up to 40-50%, the painted materials may keep the high electrical conductivity due to the high carrier concentration, the low density of pores. In order to show that these data are reliable, authors should give a theoretical result based on a porous material with the porosity up to 40-50%, not based on porous materials with the porosity less than 20%.

Response: As the reviewer suggested, we cited a **new reference (Landauer, R. Electrical Transport and Optical Properties of Inhomogeneous Media, American Institute of Physics, New York, 1978, pp. 2–45.)**. Landauer reported the numerical simulations on electrical properties of inhomogeneous system containing highly conductive and insulating components with a wide range of their portions, which can be appropriate to understand the porosity effect on the electrical properties of the painted materials because the pore sites can be considered as insulating component. The predicted electrical conductivity with the volume fraction of 45~50% in the insulating component was ~40~50% of the primary electrical conductivity. Considering that the electrical conductivities of $\text{Bi}_{0.4}\text{Sb}_{1.6}\text{Te}_{3.0}$ and $\text{Bi}_{2.0}\text{Te}_{2.7}\text{Se}_{0.3}$ with full density range $1000\sim 1500 \text{ S cm}^{-1}$, it is acceptable that the n-type and p-type painted materials exhibited $650\sim 750 \text{ S cm}^{-1}$ of the electrical conductivities.